# Evaluation of long-read sequencing for Ostreid herpesvirus type 1 genome characterization from *Magallana gigas* infected tissues

Aurélie Dotto-Maurel,[1] Camille Pelletier,[1] Lionel Degremont,[1] Serge Heurtebise,[1] Isabelle Arzul,[1] Benjamin Morga,[1] Germain Chevignon[1]

**ABSTRACT** Since the 1990s, the Pacific oyster *Magallana gigas* has faced significant mortality, which has been associated with the detection of the Ostreid Herpesvirus type 1 (OsHV-1). Due to the complex genomic architecture and the presence of multiple genomic isomers, short-read sequencing using Illumina method struggles to accurately assemble tandem and repeat regions and to identify and characterize large structural variations in the OsHV-1 genome. Third-generation sequencing technologies, as long-read real-time nanopore sequencing from Oxford Nanopore Technologies (ONT), offer new possibilities for OsHV-1 whole-genome analysis. Identification of the best method for extraction of high molecular weight (HMW) DNA and development of accurate bioinformatic pipelines for its characterization are now required. To this end, we evaluated and compared six HMW methods and one conventional DNA extraction kit for their ability to extract OsHV-1 DNA from *M. gigas*-infected tissues. We then evaluated the ability of ONT sequencing to produce an accurate OsHV-1 genome from both whole-genome and "adaptive sampling" (AS) sequencing approaches. Finally, we evaluated the efficiency of bioinformatics tools for *de novo* assembly and consensus calling to generate accurate OsHV-1 genomes. The HMW DNA extraction kit coupled with ONT sequencing and dedicated bioinformatics tools allowed us to produce accurate OsHV-1 genomes compared to those assembled using Illumina technology. The AS approach allowed up to 60% enrichment for viral data, and the long reads generated by ONT allowed the characterization of OsHV-1 isomers. Together with its portability, this sequencing shows great promise as a diagnostic tool for the characterization of unculturable aquatic viruses directly from host tissues.

**IMPORTANCE** Many aquatic viruses threaten commercially valuable species and cause significant economic losses during outbreaks. To improve our understanding of the origin, transmission patterns and spread of these viruses, additional genomic data are essential. However, genomic characterization of unculturable large DNA viruses is a major challenge. In the present study, we have successfully evaluated the ability of ONT sequencing and adaptive sequencing (AS) to sequence and assemble the complete OsHV-1 genome. Our results show that it is now possible to sequence the whole genome of large DNA viruses directly from infected host tissue, without the need for prior *in vitro* propagation or prior laboratory steps for virus enrichment.

**KEYWORDS** virus, Ostreid herpesvirus type 1, long reads, genome, Oxford Nanopore Technologies, *Magallana gigas*

Ostreid herpesvirus type 1 (OsHV-1) is a double-stranded DNA virus of the *Malacoherpesviridae* family that has been associated with mass mortality events of Pacific

**Peer Reviewers** Zsolt Csabai, University of Szeged, Szeged, Hungary; Guillaume Croville, Ecole Nationale Veterinaire de Toulouse, Toulouse, France; Umberto Rosani, Universita degli Studi di Padova, Padova, Italy

Address correspondence to Germain Chevignon, Germain.Chevignon@ifremer.fr.

The authors declare no conflict of interest.

oyster *Magallana gigas* (formerly named *Crassostrea gigas*), the most widely produced mollusk in the world (1, 2). OsHV-1 has impacted oyster production in many production basins worldwide for 15 years (3–9). It can cause mass mortalities of up to 100% within days (10, 11) and several factors trigger disease development including seawater temperature, oyster age, and genetics background (8, 11–16). While this virus cannot be cultured *in vitro*, OsHV-1 infection can be propagated *in vivo*. Therefore, its genomic characterization relies on our ability to enrich samples for viral nucleic acids.

OsHV-1 has a linear genome of approximately 207 kilobases (kb) and its genomic organization is similar to other Herpesviruses. This genome consists of two invertible unique regions $U_L$ for "Unique long" (170 kbp) and $U_S$ for "Unique short" (3.5 kbp) each flanked by tandem and inverted repeats $TR_L/IR_L$ (7.5 kbp each) around the $U_L$ and $TR_S/IR_S$ (9.7 kbp each) around the $U_S$ and separated by a third unique region named X (1.5 kbp) for a global genomic organization $TR_L$-$U_L$-$IR_L$-X-$IR_S$-$U_S$-$TR_S$. In addition, as characterized previously (1) similarly to another herpes virus (17), four majors isomers of the virus coexist at equimolar levels in the same samples: isomers P for Prototype with $U_L$ and $U_S$ in the forward orientation, $I_L$ for Inverted $U_L$, $I_S$ for Inverted $U_S$ and $I_{LS}$ for Inverted $U_L$ and $U_S$. Furthermore, each of these four major isomers has three minor isomers: $D_X$ where the X region is Duplicated (i.e., X-$TR_L$-$U_L$-$IR_L$-X-$IR_S$-$U_S$-$TR_S$), $T_X$ where the X region is Translocated at the left termini of the genome (i.e., X-$TR_L$-$U_L$-$IR_L$-$IR_S$-$U_S$-$TR_S$), and $N_X$ where the X region is absent (i.e., $TR_L$-$U_L$-$IR_L$-$IR_S$-$U_S$-$TR_S$) (1, 17, 18).

To date, detection and characterization of OsHV-1 virus in infected oysters are mainly performed by PCR and its genome is regularly sequenced using the Illumina short-read sequencing method (11, 19). However, Illumina sequencing does not allow for accurate assembly of tandem and inverted repeats (i.e., $TR_L/IR_L$ and $TR_S/IR_S$), resulting in the assembly of a Non-Redundant (NR) genome version (i.e., $U_L$-$IR_L$-X-$IR_S$-$U_S$) (19).

While the OsHV-1 life cycle is not well characterized, research perspectives for its characterization include better identification of the genomic modification of the virus during the cycle. Those modifications could include the production of a noncanonical genome with large structural variation or could involve variation in the number of genomic isomers. Currently, the characterization of such genomic alteration can only be envisioned thanks to long-read approaches.

In the past decade, new sequencing technologies allowing long DNA fragment sequencing have been developed by Oxford Nanopore Technologies (ONT) and Pacific Bioscience (PacBio). Recently, we showed that ONT sequencing allowed us to use long-read technology to assemble a complete OsHV-1 genome and accurately characterize viral isomers from TFF-filtered viral particles (18).

While the combination of TFF and ONT sequencing has shown promising results in improving the genomic characterization of OsHV-1, the implementation of this approach is complex and requires a large amount of biological material, making it difficult to use for OsHV-1 genomic characterization at the individual host level. To achieve the goal of real-time sequencing of the OsHV-1 genome at the individual host level, it is necessary to simplify the procedure and reduce the number of laboratory steps involved in sample processing.

ONT has recently introduced a selective sequencing approach named "adaptive sampling" (AS), which enables real-time enrichment of specific sequences of interest. This method is software-driven and provides the ability to accept or reject DNA molecules within individual nanopore sequencing units. By rapidly analyzing the first few hundred base pairs of a DNA molecule and comparing them to a reference sequence, it determines whether the molecule matches the target or not. If a match is found, the molecule is fully sequenced. If not, the electric current in the pore is reversed, ejecting the DNA molecule and allowing the pore to capture a new one (20) (Fig. 1).

This approach has already been successfully tested with a number of different organisms and has been able to enrich viruses, bacteria, or mammals by a factor of up to 7 (21–24).

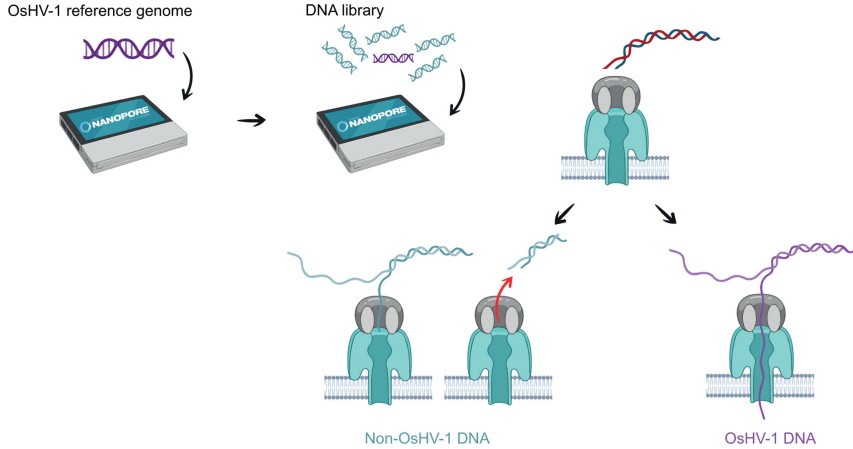

**FIG 1** Schematic representation of adaptive sampling with OsHV-1 enrichment.

This study aims to evaluate the ONT sequencing technology to characterize the OsHV-1 genome directly from infected oyster tissues. To achieve this, we used infected moribund oysters to first evaluate a number of high molecular weight (HMW) DNA extraction kits for extracting OsHV-1 DNA directly from infected host tissues. We then tested the ability of ONT to sequence the OsHV-1 genome from DNA extracted from a single host individual with and without AS. To determine the level of accuracy, the ONT genome was compared to a genome generated using Illumina technology, and we determined the minimum number of ONT reads required to obtain an ONT OsHV-1 genome with 99.99% accuracy. Finally, we evaluated the quality of different consensus callers for producing accurate OsHV-1 genomes without the time-consuming *de novo* assembly steps.

## RESULTS

### Yield, quality metrics, and OsHV-1 copy number of extracted DNA

Our results showed that the majority of the kits tested on 20 ± 2 mg of tissues yielded similar amounts of DNA ranging from 10.6 to 14.9 µg, except the MagAttract and Zymo kits, which yielded an average of 8.7 and 4.6 µg, respectively (Fig. 2A). Most of the kits tested produced a consistent yield across different samples, except the Circulomics kit, which had a yield range from 3.7 µg to 27.5 µg.

The extracted DNA was then migrated onto an agarose gel to assess the DNA size distribution of the fragments for each extraction method. Samples with the longest DNA fragment sizes were those extracted using the Monarch, Circulomics, and Macherey Nagel kits, while the QiAgen kit produced predominantly fragmented DNA, as expected. Promega, Zymo, and MagAttract kits produced similarly mixed DNA fragments, with some long fragments and some fragmented DNA (Fig. 2B).

The DNA yields and qualities claimed by the manufacturers are shown in Table 1. Overall, we obtained DNA yields that were in line with the claims, except for the Promega kit which yielded 10 times more DNA than expected.Table 1 On the other hand, we extracted less DNA than expected from the MagAttract and the Macherey Nagel kits. As expected, based on the manufacturer's claims, the Monarch, Circulomics, and Macherey Nagel kits provided the longest apparent DNA fragments (Table 1).

The number of OsHV-1 genome copies per ng of DNA was similar among kits except for the Macherey Nagel kit ranging from 115,000 to 7,520,000 copies per ng of DNA (Fig. 2C), indicating that all kits have similar abilities to extract OsHV-1 DNA from infected oyster tissue.

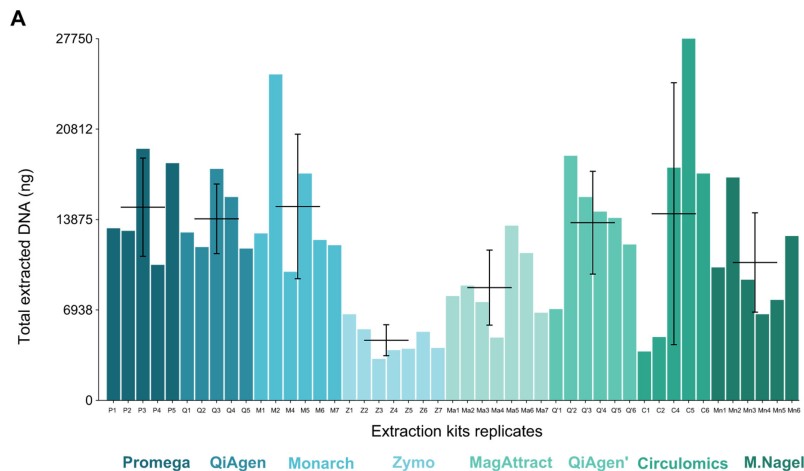

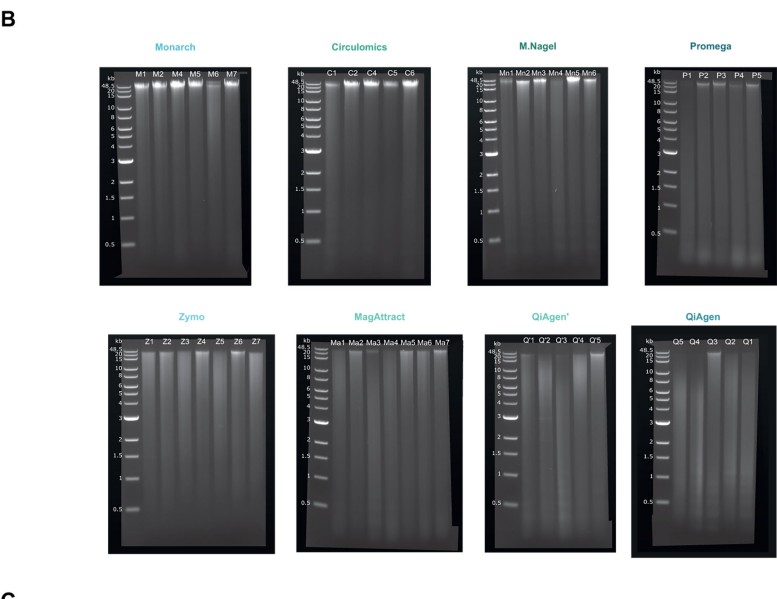

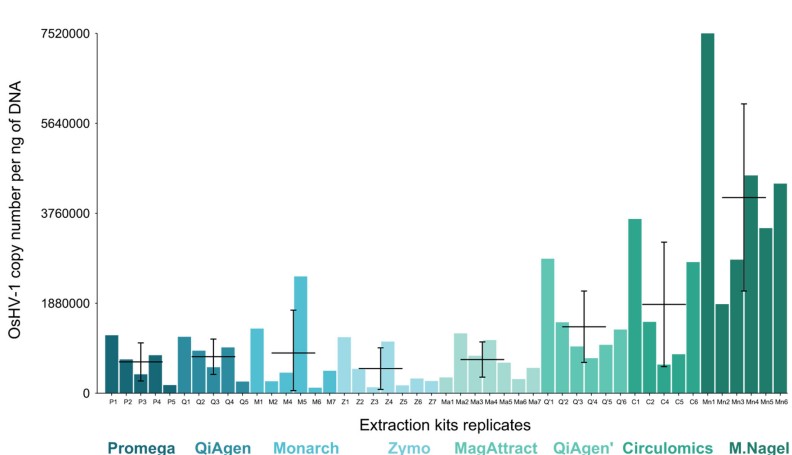

**FIG 2** DNA extraction characteristics. (A) Total extracted DNA in ng was obtained with the seven different HMW DNA extraction kits. The X-axis represents the replicates of the extraction kits (five replicates for Promega and QiAgen; six replicates for QiAgen', Circulomics, and Macherey Nagel; and

**Fig 2 (Continued)**

seven replicates for Monarch, Zymo, and MagAttract) while the Y-axis represents the total DNA extracted (ng). The horizontal black lines correspond to the mean of extracted DNA for each extraction kit while the vertical black lines represent its standard deviation. (B) Agarose gel electrophoresis of genomic DNA isolated from OsHV-1-infected oyster tissue using seven different HMW DNA extraction kits. The first line of each gel corresponds to DNA ladders ranging from 0.5 to 48.5 kb. (C) OsHV-1 genome copy number per ng of extracted DNA for seven different HMW DNA extraction kits. The X-axis represents the replicates of the extraction kits (five replicates for Promega and QiAgen; six replicates for QiAgen′, Circulomics, and Macherey Nagel; and seven replicates for Monarch, Zymo, and MagAttract) while the Y-axis represents the OsHV-1 copy number per ng of DNA. The horizontal black lines correspond to the mean of extracted DNA for each extraction kit while the vertical black lines represent its standard deviation.

## ONT allows sequencing of OsHV-1 from infected oyster tissue

### Adaptive sampling enables the enrichment of OsHV-1 reads

The six ONT sequencing runs produced a total of 13.4 M reads and 18.6 G bases, ranging from 0.59 M reads for run 4 to 5 M reads for run 2, and from 0.3 G bases for run 4 to 8.4 G bases for run 2 (Fig. S1).

By mapping the reads against the *M. gigas* and OsHV-1 µVarA reference genomes, we were able to determine the origin of the reads for each DNA sample in each sequencing run (Fig. 3). Overall, we obtained between 1.19% and 17.78 % of OsHV-1 reads, between 76.78% and 95.94% of *M. gigas* reads, between 0.01% and 2.11 % of reads that aligned to both the OsHV-1 and *M. gigas* genomes, and between 1.38% and 6.39 % of reads that did not align to the OsHV-1 genome neither to and *M. gigas* genome.

Run 1, which used DNA samples with a higher amount of OsHV-1 per ng of extracted DNA, had the highest level of OsHV-1 reads among the "whole sequencing" runs (runs 1, 2, and 3). For run 1, the proportion of OsHV-1 reads ranged from 4.4% for the MagAttract kit to 17.8% for the Macherey Nagel kit, with an average of 9.7% (Fig. 3). Sequencing runs 2 and 3, which used samples selected for their higher apparent fragment size on agarose gels, had generally a lower proportion of OsHV-1 compared to run 1 (Fig. 3). In addition, the proportions of OsHV-1 reads and bases were very similar for each sample within the three "whole sequencing" runs. Conversely, the three sequencing runs where AS was enabled for OsHV-1 enrichment showed at least a twofold increase in the percentage of OsHV-1 bases sequenced compared to the percentage of reads sequenced (Fig. 3). For run 4, where DNA fragments were size selected, there was an even more pronounced enrichment of OsHV-1 bases compared to the number of reads, with Monarch, Macherey Nagel, and Circulomics DNA samples containing up to 68.1%, 56.3%, and 38.9% of OsHV-1 sequenced bases, respectively (Fig. 3). *M. gigas* reads had higher N50 values for most of the sequenced samples in the three "whole sequencing" runs, while reads that did not align with *M. gigas* or OsHV-1 had the lowest N50 value. OsHV-1 N50 values were always close to those of *M. gigas*, but rarely higher (Macherey Nagel and QiAgen samples from run 1) (Fig. 4). However, in the AS runs, non-OsHV-1 reads (i.e., *M. gigas* and other reads) showed a significant decrease in N50 values with values around 500 bp.

**TABLE 1** Expected and obtained DNA yield and fragment size by the different DNA extraction kits

| Kits | Expected | | *M. gigas* tissue | |
| --- | --- | --- | --- | --- |
| | DNA yield | DNA fragment size | DNA yield | DNA fragment estimation |
| Promega | 17 µg for Spinach tissue | 64% > 50 kpb | 10–20 µg | >48.5 kb -- |
| Monarch | 89–20.8 µg | 50 to >500 kpb | 9–25 µg | >48.5 kb ++ |
| QiAgen | 5–30 µg | Up to 50 kpb | 7–19 µg | >48.5 kb -- |
| Zymo | 10 µg per 50 µL of beads | Up to 150 kpb | 3–7 µg | >48.5 kb + |
| MagAttract | 12–70 µg | Up to 150 kpb | 4–13 µg | >48.5 kb + |
| Circulomics | 5–100 µg | 50 to >300 kpb | 3–28 µg | >48.5 kb ++ |
| Macherey Nagel | 20 µg | 0,5 to >300 kpb | 6–17 µg | >48.5 kb ++ |

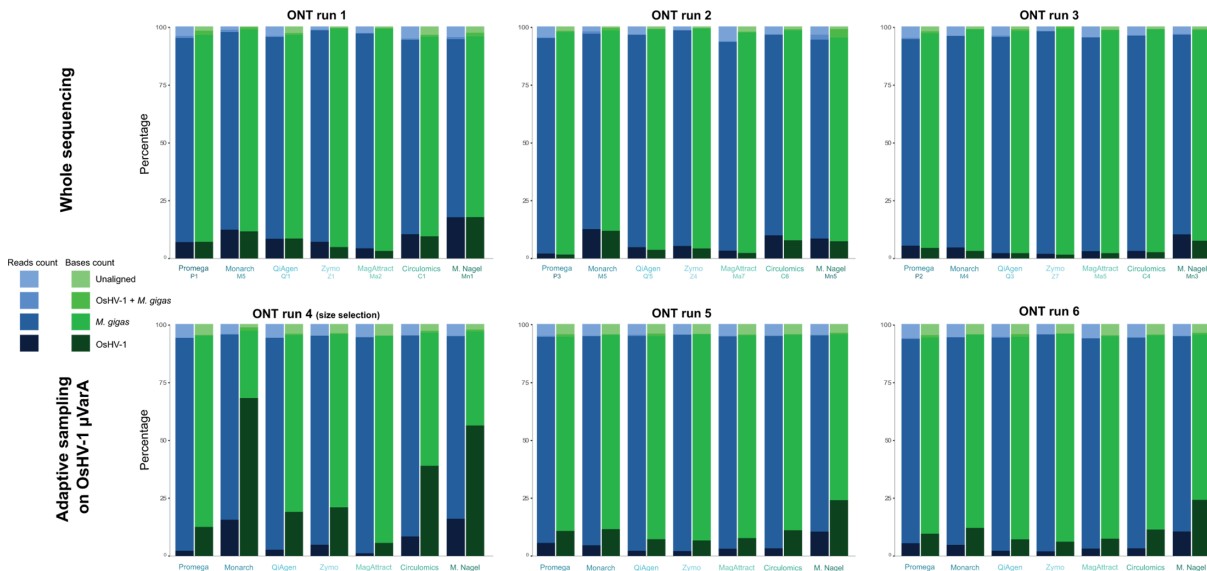

**FIG 3** Classification of sequenced reads and bases for six ONT sequencing runs and seven different HMW DNA extraction kits. Histograms showing the proportion of reads (blue) and bases (green) for OsHV-1 (dark colors), *M. gigas* (mid-dark colors), "OsHV-1 +*M. gigas*" (mid-light colors), and "Unaligned" (light colors) for six sequencing run and seven different HMW DNA extraction kits. "Unaligned" corresponds to reads that did not align to either OsHV-1 µVarA or *M. gigas*. "OsHV-1 +*M. gigas*" corresponds to reads that align on OsHV-1 µVarA and on *M. gigas*.

Furthermore, OsHV-1 reads in runs 5 (OsHV-1 N50 = 2,065 bp) and 6 (OsHV-1 N50 = 2,302 bp) did not show an increase in N50 compared to runs without AS (run 1 OsHV-1 N50 = 4,436 bp, run 2 OsHV-1 N50 = 2,627 bp and run 3 OsHV-1 N50 = 2,267 bp), but OsHV-1 N50 values benefited greatly from the combination of the AS and size selection. Indeed, in run 4, N50 values ranged from 4,534 bp for the MagAttract sample to 8,350 bp for the Monarch sample, with an average OsHV-1 N50 of 5,820 bp (Fig. 4).

## ONT sequencing allows accurate characterization of the complete OsHV-1 genome

### Illumina OsHV-1 genome assembly

In parallel, tissues infected with the same OsHV-1 strain were sequenced using Illumina technology to provide a comparative basis for the evaluation of the ONT technology. The Illumina sequencing run yielded more than 102 M of 150 bp paired-end reads, distributed as follows: 0.90% of OsHV-1 reads, 93.82% of *M. gigas* reads, and 5.26% of unaligned reads (Fig. 5 top panel). Our pipeline, based on the *de novo* assembler SPAdes, produced five OsHV-1 contigs ranging from 2,484 to 164,393 bp (Fig. 5, top panel). These five contigs were then manually assembled to construct an OsHV-1 NR genome in which only one copy of each repeat was conserved ($U_L$-$IR_L$-X-$IR_S$-$U_S$, see Materials and Methods for more details). By re-aligning the Illumina reads on the new OsHV-1 NR genome, we were able to correct the junction between contigs C7709 and C3973 (addition of one A/T bp) and to confirm that the other metaSPAdes contigs were correctly assembled. As expected, since the NR genome contains only one copy of these duplicated regions, we observed that the read coverage within $IR_L$ and $IR_S$ was twice that of the unique regions $U_L$, $U_S$, and X (Fig. 6, top panel).

### ONT OsHV-1 genome assembly

Shasta produced the smallest contigs (N50: 3 kb) and the highest number of OsHV-1 and non-OsHV-1 contigs with a total of 494/1,314 and 494/1,454 contigs for all reads and the top 10% longest read data set, respectively. Smartdenovo struggled to assemble a complete OsHV-1 genome for all data sets except with the top 0.1% longest reads (N50:

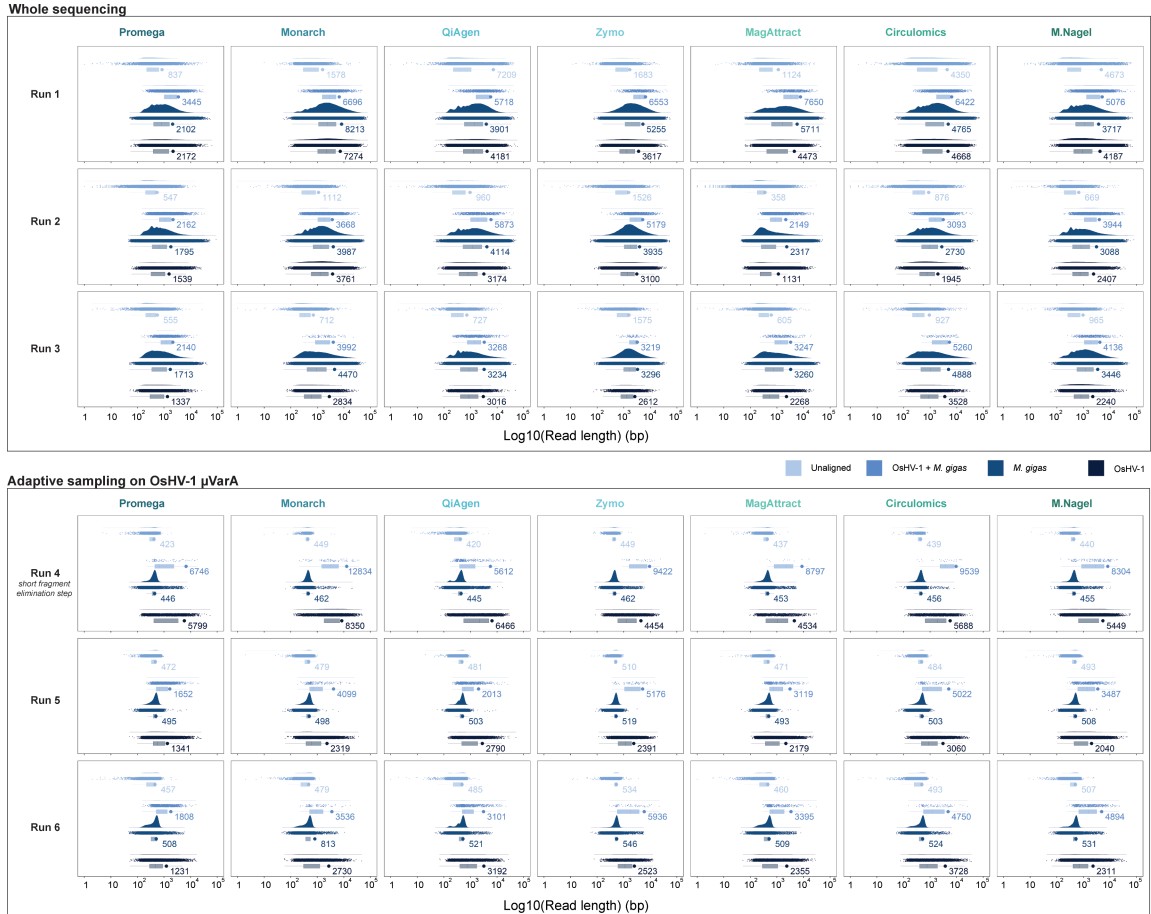

**FIG 4** Raincloud plot of read length distribution for six ONT sequencing runs and seven different HMW DNA extraction kits. Each of the 42 graphs is shown on the x-axis: the reads density distribution, the reads length distribution as a dot plot, and the bar plot representing log10 (read length) in bp. Different colors correspond to OsHV-1 reads (dark blue), *M. gigas* reads (mid-dark blue), "OsHV-1 +*M. gigas*" reads (mid-light blue), and "Unaligned" reads (light blue). Colored dots on the bar plot represent N50 values. "Unaligned" corresponds to reads that did not align to either OsHV-1 µVarA or *M. gigas*. "OsHV-1 +*M. gigas*" corresponds to reads that align on OsHV-1 µVarA and *M. gigas*.

237 kb and 1/1 contig). RA required a low coverage to assemble the OsHV-1 genome and generated an extended OsHV-1 contig of 233 kb for the top 0.1% longest reads data set only. Raven generated an extended OsHV-1 contig of more than 200 kb for all but the "all read" data set, however, like Shasta, never fully assembled the $U_L$ region. Flye produced the highest number of long contigs in both the OsHV-1 and the non-OsHV-1 data sets, with a total of 58/32, 28/4, 3/4, and 6 contigs and N50 of 44 kb, 32 kb, 179 kb, and 186 kb, respectively. Canu produced 1 to 4 contigs among the data sets and seemed to be the best tool for OsHV-1 assembly. In addition, Flye and Canu consistently produced one OsHV-1 contig that was longer than 170 and 228 kb, respectively, across all data sets. This contig is always shorter than expected for Flye, whereas it is always longer than expected for Canu.

### Illumina short-read and ONT long-read sequencing produce nearly identical genomes

Pairwise alignment of the Illumina and ONT assembled genomes revealed only 15 bp differences between the two genomes, corresponding to 99.992% average nucleotide identity (ANI). Interestingly, all of these differences occurred only in G/C homopolymers, which is a known limitation of long-read sequencing and more specifically of ONT sequencing (Fig. 7).

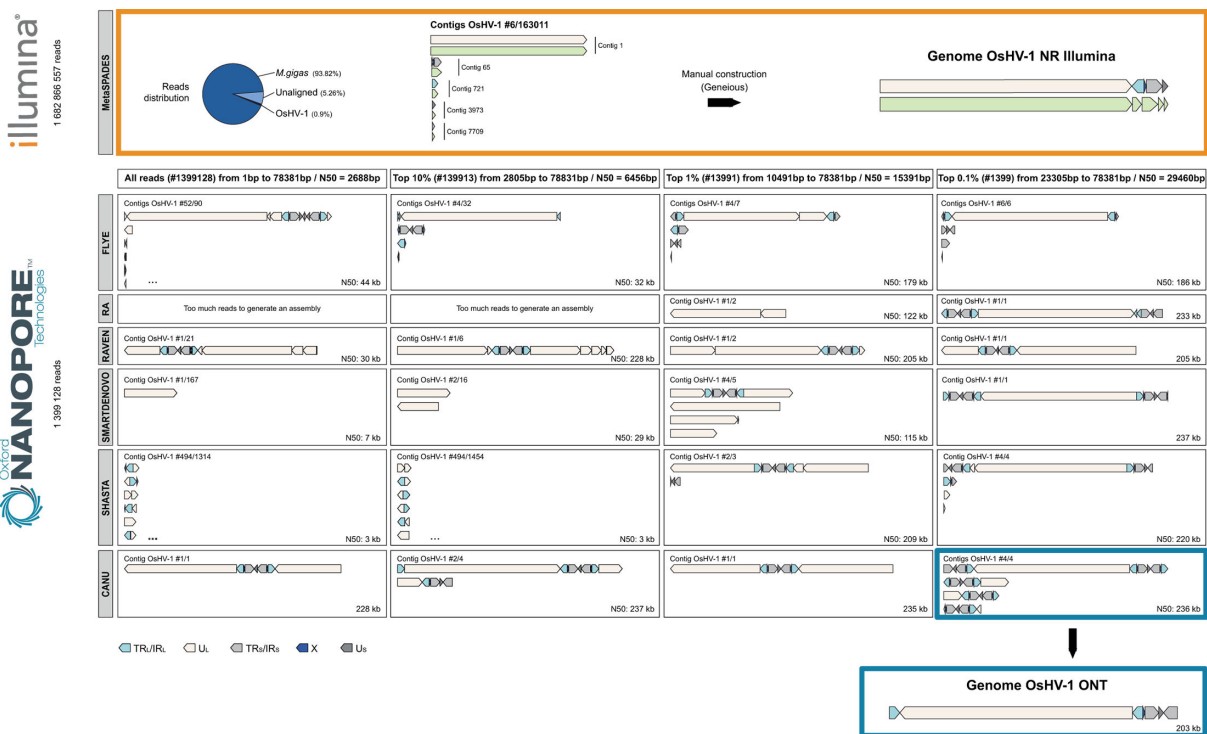

**FIG 5** Evaluation of *de novo* assembly tools for short- and long-read sequences. Illumina assembly: The distribution of the reads used for the assembly is shown on the pie chart with *M. gigas* reads in dark blue, Unaligned reads in light blue, and OsHV-1 reads in black. MetaSPADES was used to assemble an OsHV-1 reference genome using Illumina short reads. Five contigs were generated and manually scaffolded to produce a non-redundant OsHV-1 genome. ONT assembly: Six different long-read *de novo* assemblers (Flye RA, Raven, Smartdenovo, Shasta, and Canu) were evaluated with four different read sets: all non-*M. gigas* reads from all sequenced samples from all six ONT sequencing runs; the top 10%, 1%, and 0,1% longest non-*M. gigas* reads from all sequenced samples from all six ONT sequencing runs. This evaluation produced 24 panels showing the contigs produced by each *de novo* assembler and each data set. Only the OsHV-1 contigs are visualized and each contig has been annotated for specific OsHV-1 regions: TR$_L$/IR$_L$ (turquoise); U$_L$ (white); X (blue); TR$_S$/IR$_S$ (gray); U$_S$ (dark gray).

We then evaluated the minimum ONT read depth required to assemble and polish an OsHV-1 genome as accurately as possible (i.e., to achieve the highest ANI compared to the Illumina genome). The polishing step increased the ANI from 99.947% for the Canu assembly alone to 99.994% for the Medaka-polished genome (Fig. 8). Finally, we found that when the coverage of the OsHV-1 genome reached 50 reads, the quality of the Canu assembly polished with a run of Medaka was close to the Illumina genome (99.994% accuracy) (Fig. 8).

## ONT consensus callers show contrasting results depending on read depth

To ensure a rapid and accurate capture of OsHV-1 genomes, we can also call a consensus sequence directly after aligning the reads to a reference genome. This approach bypasses the need for *de novo* assembly and polishing steps, which can be time-consuming.

To evaluate this approach, we tested reference mapping with reads coming from all 42 libraries individually, with pooled reads from each of the six runs, and with a pool of all the sequenced reads on the new *de novo* assembled and polished ONT OsHV-1 genome. These 49 libraries were consensus called using five different tools (See Materials and Methods for more details). For each consensus call, we obtained 49 consensus genomes and compared them pairwisely using the alignment *all versus all* (Fig. S2). Since all 42 ONT sequenced libraries were prepared with DNA extracted from tissues infected by the same viral inoculum, the observed ANI is mainly the result of ONT sequencing error combined with the ability of the different consensus callers to correct this error. Our results show that Clair3 and Pepper DeepVariant and to a lesser extent Medaka were the

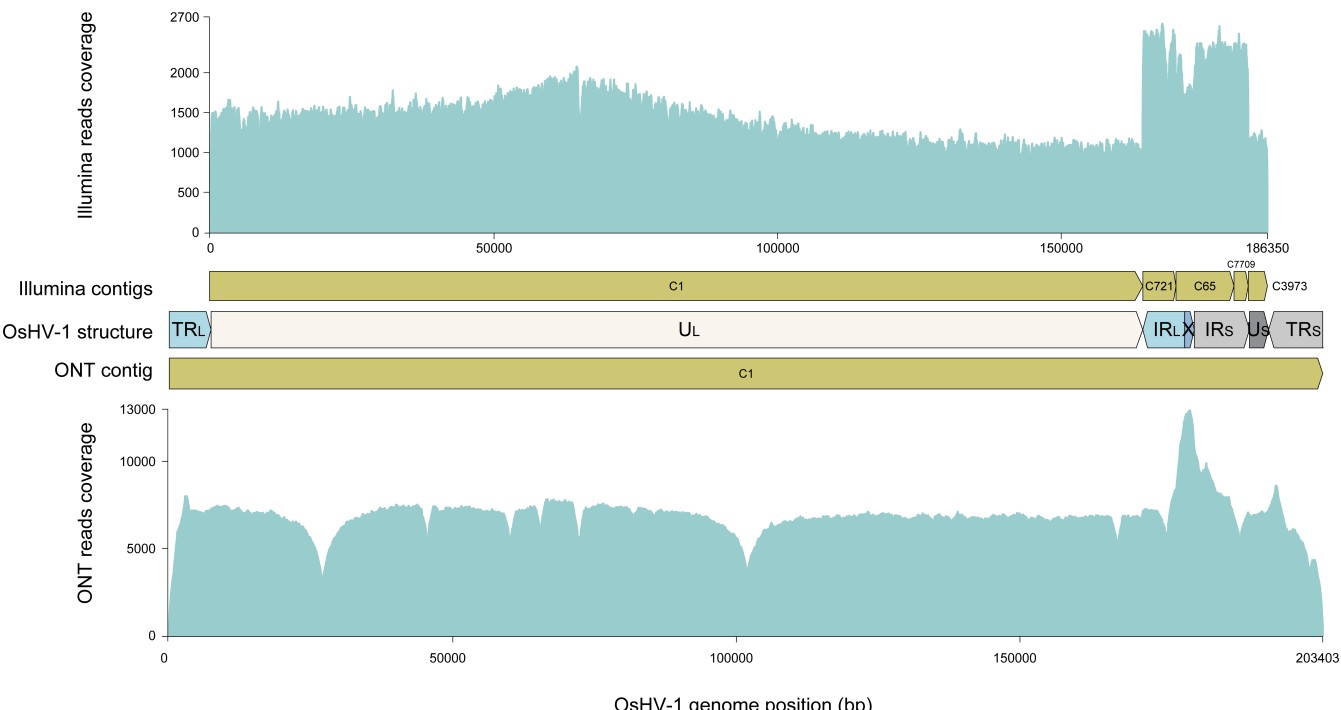

**FIG 6** Distribution of read coverage on the Illumina and ONT assembled genome. Top panel: Distribution of Illumina reads (turquoise) on Illumina assembled contigs (yellow). Bottom panel: Distribution of ONT reads (turquoise) on the ONT-assembled contig (yellow). The genomic architecture of OsHV-1 is shown with the specific regions: $TR_L/IR_L$ (turquoise); $U_L$ (white); X (blue); $TR_S/IR_S$ (gray); and $U_S$ (dark gray). The Y-axis represents the reads coverage while the X-axis represents the position of the OsHV-1 genome.

most consistent for producing accurate genomes (i.e., producing the genome closest to the one generated from Illumina) (Fig. S2). When comparing the 49 ONT consensus calls to the Illumina assembled genome, Medaka, Clair3, and Pepper DeepVariant had an ANI of 99.991% with the Illumina genome (Fig. 9). However, Pepper DeepVariant and Clair3 were more consistent than Medaka to produce accurate genomes across all coverage levels tested (Fig. 9). Longshot performed best at generating accurate genomes with coverage greater than 100 (Fig. S2; Fig. 9), achieving an ANI greater than 99.990% for read depths ranging from 104 to 6,697 (Fig. 9). Finally, Nanocaller excelled at producing accurate genomes with coverage below 200 (Fig. 9; Fig. S2), achieving an ANI greater than 99.990% for a coverage between 11 and 244 (Fig. 9).

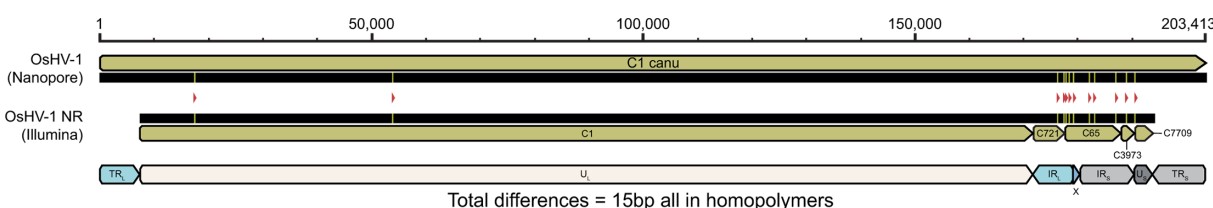

Total differences = 15bp all in homopolymers

**FIG 7** Pairwise alignment of ONT and Illumina assembled genomes. Alignment of the Illumina NR OsHV-1 genome (GenBank: GCA_964656475.1). generated by MetaSPADES with the non-*M. gigas* Illumina reads against the ONT OsHV-1 genome (GenBank: GCA_964656485.1) generated by Canu with the 0.1% longest non-*M. gigas* ONT reads. These two genomes differ by a total of 15 SNPs compared to a consensus of 186 kb, all located in homopolymers (red triangles). Yellow bars represent the OsHV-1 genome position of these SNPs. Light green bars represent contigs used to generate references and black bars represent nucleic sequences. The OsHV-1 structure is shown with the specific regions: $TR_L/IR_L$ (turquoise); $U_L$ (white); X (blue); $TR_S/IR_S$ (gray); and $U_S$ (dark gray).

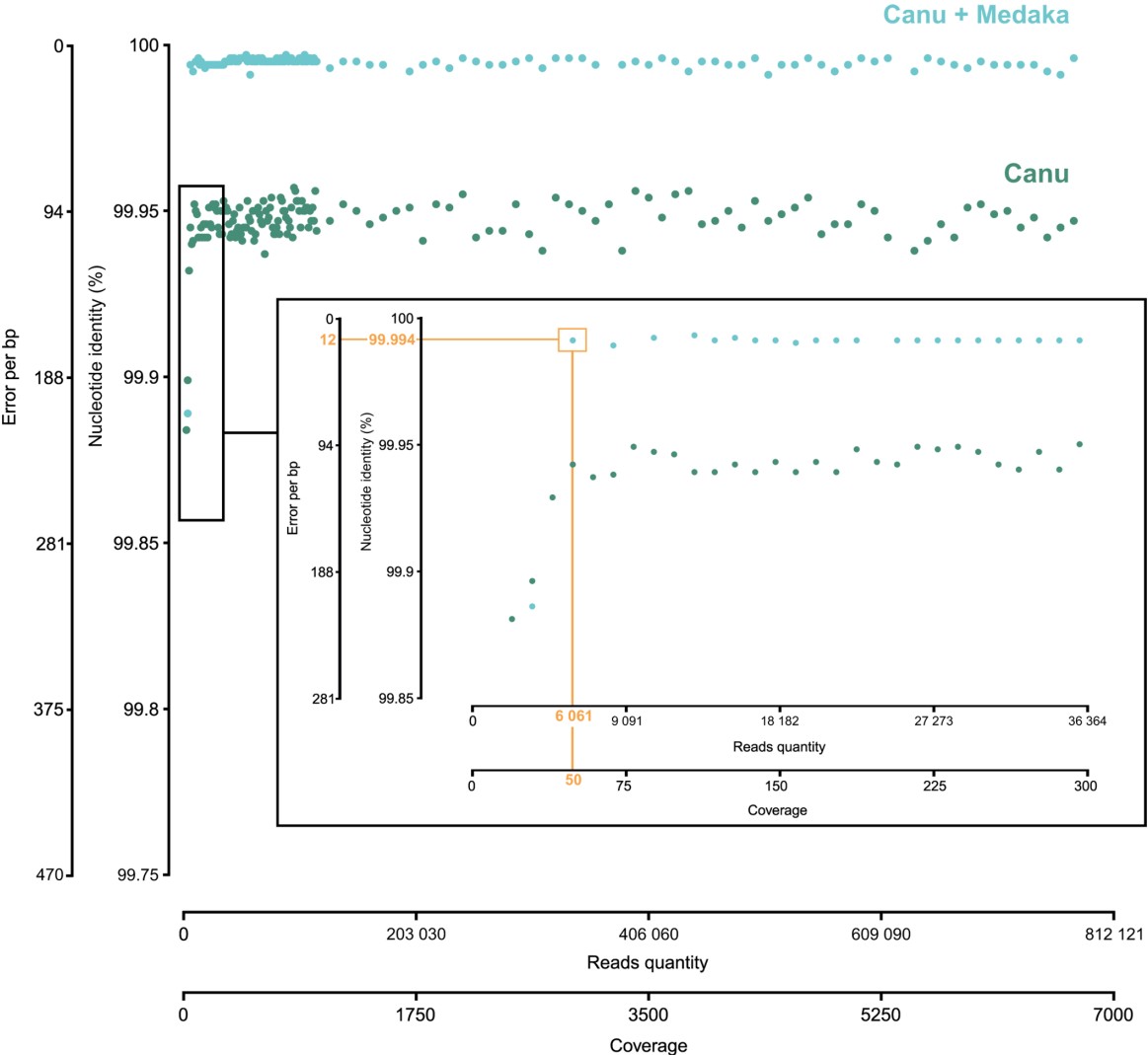

**FIG 8** Evaluation of the minimum ONT read depth required to assemble an accurate OsHV-1 genome. Dot plots of nucleotide identity between the assembled genomes and the Illumina OsHV-1 genome according to coverage. Each blue dot represents an ONT OsHV-1 genome assembled with Canu but not polished. Each green dot represents an ONT OsHV-1 genome assembled with Canu and polished with Medaka. Nucleotide identity (%) and the error per bp between ONT and Illumina OsHV-1 genomes are shown on Y-axes. The number of reads used for the assembly and the corresponding coverage on the Illumina genome are shown on X-axes. The center panel is an extension of low coverage from 0 to 300. The orange rectangle frames the first ONT assembled and polished genome which is accurate (99.994% nucleotide identity between ONT and Illumina genome) and was generated with 6,061 OsHV-1 reads.

## ONT sequencing allows the detection of OsHV-1 structural variation and the quantification of the four major isomers

The Canu assembly of the top 0.1% longest reads provided the most reliable genome structure and produced three additional contigs corresponding to the other three major OsHV-1 isomers (Fig. 5 bottom panel) (1). Similar to our previous results (18), we quantified the relative abundance of the four isomers by screening all the reads long enough to cover at least a part of both the $U_L$ and the $U_S$ regions. Thus, we identified 137 long reads divided into 32 $I_{LS}$ isomers, 36 P isomers, 39 $I_L$ isomers and 30 $I_S$ isomers (Fig. 10).

Furthermore, thanks to the ability of ONT to generate long reads, we use Sniffle to analyze the structural variations (SV) directly within the OsHV-1 reads. We detected 18 SVs, 6 of which were likely to be related to the $U_L$ inversion. As the reads were shorter than the $U_L$ length (<164 kb), we could not determine whether they were true SVs or

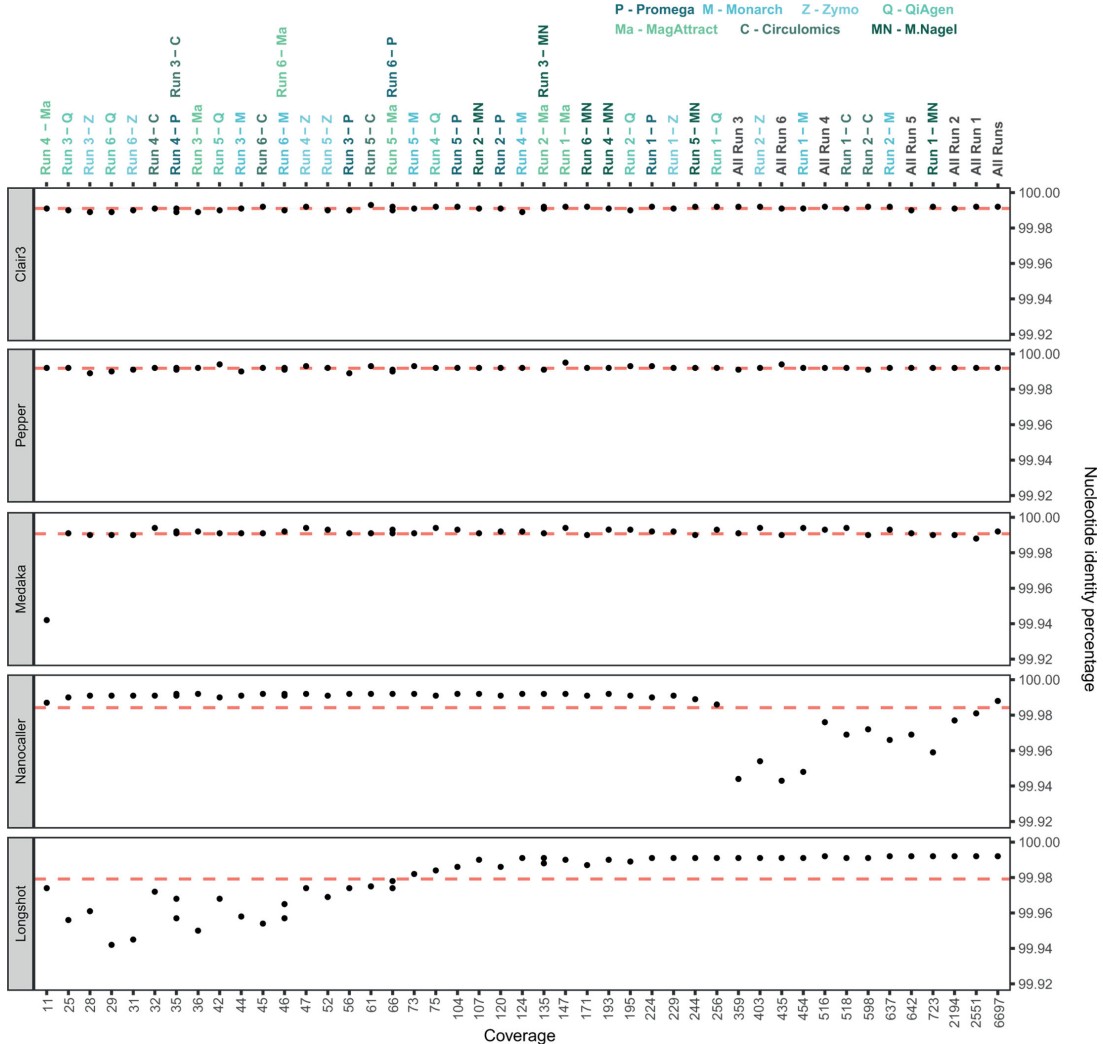

**FIG 9** Consensus caller scoring. Pairwise comparison of OsHV-1 consensus sequences generated by five different long reads consensus callers: Clair3, Pepper DeepVariant, Medaka, Nanocaller, and Longshot. Red lines represent the mean percentage of identity for each of the consensus caller tested. The Y-axis represents the percentage of nucleotide identity and the X-axis represents read coverage.

artifacts due to the OsHV-1 isomers, so we decided to filter them out. Finally, we obtained 12 SVs (Fig. 11), including a deletion of 1,510 bp corresponding to the absence of the X region supported by 91 reads, an inversion of 3,371 bp corresponding to the inversion of the $U_S$ region supported by 46 reads, and a duplication of 1,515 bp located in the X region, corresponding to the duplication of at least two copies of the X region supported by 12 reads. We also detected 9 SVs spread across the $U_L$, $IR_L$, X, $IR_S$, $U_S$, and $TR_S$ regions, each supported by only two reads. Due to the low number of reads supporting these SVs, they were considered spurious. Six of them were located in the $U_L$ region; one overlapped the $U_L$ and $IR_L$-X regions; one was located over two parts of the $U_L$ region and in the $U_S$ region; and the last SV corresponds to a deletion of approximately 8,200 bp that includes the entire $U_S$ region and part of the $TR_S$ and $IR_S$ regions (data not shown).

To determine the distribution of reads supporting the different SV at the individual library level, we analyzed each DNA extraction from each kit individually for each sequencing run, specifically for the X deletion, $U_S$ inversion, and X duplication identified from the whole data set (Fig. S3). The $U_S$ inversion was identified in 35 reads for run 1; two for run 2, eight for run 4, and one for run 6. Of these, 22 were generated from the Monarch kit, 2 from the Promega kit, 1 from the QiAgen kit, 5 from the Circulomics kit,

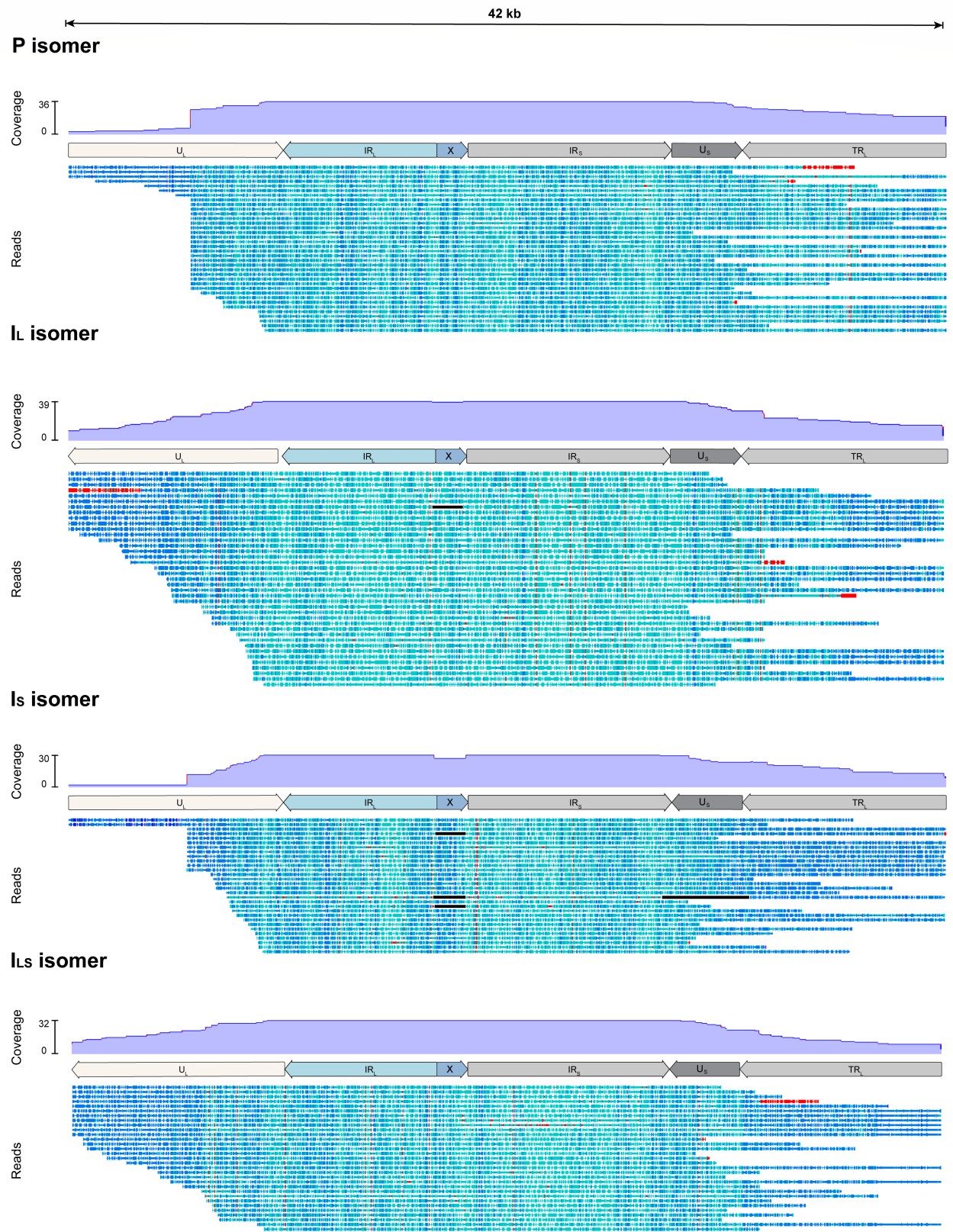

**FIG 10** OsHV-1 isomer characterization. Coverage distribution of long reads (>42 kb) overlapping at least part of $U_L$ and $U_S$ with a mapping quality score >40 against the four major isomers (P, $I_L$, $I_S$, and $I_{LS}$). Reads are color-coded at the base level by similarity from dark blue (100% identity) to red (less than 60% identity).

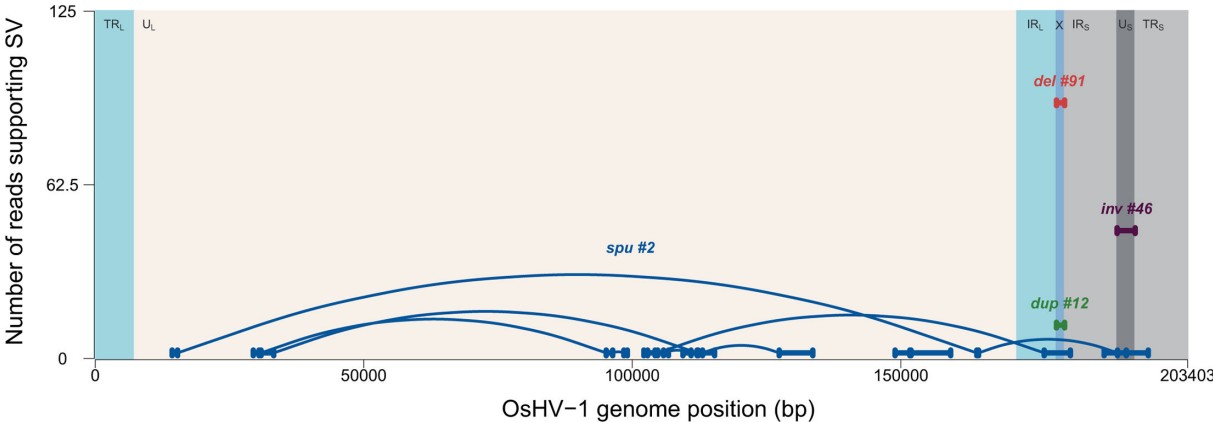

**FIG 11** Structural variation. Representation of the 12 structural variations (the X deletion [red], the U$_S$ inversion [purple], the X duplication [green], and the spurious structural variations [blue]) identified by Sniffles2 with all non-*M. gigas* reads. The genomic architecture of OsHV-1 is shown with the specific regions: TR$_L$/IR$_L$ (turquoise); U$_L$ (white); X (blue); TR$_S$/IR$_S$ (gray); and U$_S$ (dark gray). The Y-axis represents the number of reads supporting SV and the X-axis represents the position of the OsHV-1 genome.

and 16 from the Macherey Nagel kit. The X deletion was identified in 32 reads for run 1; 37 for run 2; six for run 3; five for runs 4 and 5, and six for run 6. Of these, two were generated by the Promega kit, 33 by the Monarch kit, 10 by the QiAgen kit, 19 by the Zymo kit, 5 by the MagAttract kit, 13 by the Circulomics kit, and 9 by the Macherey Nagel kit. Finally, the X duplication was identified in six reads for run 1, four reads for run 2 and 4 reads for run 6. Among them, one was generated by the Promega and the QiAgen kits, five by the Monarch kit, one by the Circulomics kit, and four by the Macherey Nagel kit.

Most of these SV-containing reads came from the first two runs (37 of the 46 reads identified as U$_S$ inversion, 69 of the 91 reads identified as X deletion, and 10 of the 12 reads identified as X duplication).

Regarding the different kits, 40% of the 149 SV-bearing reads were from the Monarch extraction, 20% from the Macherey Nagel extraction, and 13% from the Circulomics extraction.

## Virome characterization

For Illumina data, the *de novo* assembly of the 5,089,165 non-OsHV-1 and non-oyster reads produced 211,233 contigs ranging from 55 to 19,090 pb. Among those contigs, 28,503 were longer than 1 kb, 18 contained at least one viral gene, 18 contained no viral gene but had a score greater than 0.95, and 25 contigs contained neither viral nor host genes (Table S1). Of these 61 contigs, four were identified as dsDNA phages, 24 as *Lavidaviridae* virophages, 20 as ssDNA viruses and 13 as NucleoCytoplasmic Large DNA Viruses (NCLDV) (Table S1).

The ONT *de novo* assembly of the 576,052 non-OsHV-1 and non-oyster reads produced 1,039 contigs ranging in size from 228 to 57,109 bp. Among those contigs, 1,031 contigs were longer than 1 kb, 3 contained at least one viral gene, 4 contained no viral gene but had a score greater than 0.95, and 15 contigs contained neither viral nor host genes (Table S1). Of these 21 contigs, 3 were identified as dsDNA phages, 10 as *Lavidaviridae* virophages, 4 as ssDNA viruses, and 4 as NucleoCytoplasmic Large DNA Viruses (NCLDV).

## DISCUSSION

Recently, whole-genome sequencing has become a standard method to characterize and study viruses (25–27). However, this approach is still challenging when working with DNA viruses with large and complex genomes such as Herpesviruses. The inverted repeat regions in the OsHV-1 genome increase the difficulty of obtaining a properly assembled

genome. In addition, several isomers have been characterized and recently observed and quantified from purified OsHV-1 particles (1, 18). Thus, these features further complicate genome assembly using a short-read sequencing approach. Finally, because OsHV-1 is unculturable, obtaining sufficient viral material to analyze and assemble an accurate OsHV-1 genome is a real challenge.

In a previous study, we showed that ONT sequencing of viral OsHV-1 particles purified by TFF allowed us to overcome these difficulties (18). Indeed, we were able to generate up to 60% of viral bases, which represents a significant enrichment compared to previous work where the ratio of OsHV-1 bases from short-read sequencing of infected tissues ranged from 0.1% to 13.1% (19, 28, 29). However, although TFF is a powerful tool for purifying unculturable viruses from clinical and environmental isolates, it still requires additional experiments and increases cost, which slows down and complicates viral pathogen identification. Therefore, TFF cannot be used routinely.

In our study, seven DNA extraction kits were tested. Monarch, Circulomics, and Macherey Nagel kits were those that extracted the largest DNA fragment sizes. Overall, the amount of the OsHV-1 genome copy number was similar among the kits tested and was also similar to those observed in previous studies (30, 31).

Two types of sequencing were then performed: whole-genome (WG) sequencing without enrichment, and adaptive sampling (AS) sequencing with real-time enrichment for OsHV-1 sequences.

The six sequencing runs generated between 0.3 and 8.4 Gb, with the number of pores available at the start of each run ranging from 139 to 1,488. Notably, the amount of data generated did not correlate with the initial flow cell check results but seems to correlate more with the sequencing duration (Fig. S1). This is probably because the flow cell check only provides a snapshot of the number of active pores and does not account for variations in pore lifetimes. In addition, differences in pore data output rates, DNA quality, and sequencing conditions can affect pore activity and ultimately the overall data yield over time.

While WG sequencing resulted in an average OsHV-1 base count of 9.7% under all conditions, similar to previous studies where no enrichment was performed (19, 28, 29), the AS sequencing enriched our data in viral bases by at least twofold, reaching up to 60% of viral bases for some samples, approaching the results obtained with our TFF-based OsHV-1 purification (18). The levels of enrichment observed with AS were also similar to those obtained for enrichment of the human adenovirus (HAdV) where 0.24- to 7.08-fold enrichment was obtained (22). Although the availability of a reference genome is required to perform AS, this approach could overcome the limitations associated with traditional sequencing methods, such as the need for greater sequencing depth, thus reducing the cost of sequencing and data storage. In addition, up to 2% of all reads generated aligned to both OsHV-1 and *M. gigas* genome. As this is the lytic phase and there is no evidence of viral integration, these reads are likely to be artifacts resulting from the side effect of the ligase used during the ONT library preparation.

Interestingly, the N50 read length of all reads was lower than expected based on gel migration analysis, averaging approximately 2,800 bp. This size discrepancy is likely due to two main factors: (i) DNA fragmentation during the library preparation process and (ii) a sequencing bias favoring shorter DNA fragments, which pass through the pores more efficiently than longer fragments reducing the average N50 of the sequenced reads.

Improvements in the extraction of viral data from infected tissues represent a significant advance. However, optimal use of this method requires the ability to construct high-quality viral genomes. To date, complex viral genomes, such as herpesviruses, require a combination of long- and short-read sequencing for accuracy and resolution of inverted repeat regions (32, 33). Our recent validation of the accuracy of ONT sequencing in generating OsHV-1 genomes without hybrid *de novo* assembly encouraged us to evaluate its ability in *de novo* assembly directly from infected *M. gigas* tissues (18).

To our knowledge, only Canu and Flye have been tested to assemble OsHV-1 genomes from ONT reads (18). Here, we have also evaluated RA, Shasta, Smartdenovo,

and Raven to identify the best tool for OsHV-1 genome assembly. Interestingly, none of the *de novo* assemblers/data subset combinations produced a perfect OsHV-1 genome with a structure similar to the OsHV-1 reference genome (1). All tools struggled with the placement of the inverted repeat regions and with the orientation of the $U_L/U_S$ isomer inversions. However, similar to our previous results (18), Canu produced the best assembly by combining the top 0.1% longest reads. Thanks to a limited manual curation (the removal of an extra $TR_S$-$U_S$-$IR_S$-X region at the 5′ end of the genome and an extra X-$TR_L$ region at the 3′ end), we obtained a complete OsHV-1 genome with an architecture similar to the OsHV-1 reference genome (1). In addition, Canu was able to assemble three other contigs corresponding to the three other major OsHV-1 isomers (1, 18).

In our study, we compared the *de novo* ONT assembly with the Illumina short-read assembly to assess the accuracy of the ONT genome. Due to limitations in the resolution of inverted repeats and viral isomerization, the short-read approach produced five contigs that required extensive manual scaffolding to obtain an NR OsHV-1 genome structure similar to the reference genome (1, 18). While a previous comparison identified 187 bp differences between Illumina and ONT sequencing approaches (18), our present study found only 15 bp differences between the two sequencing methods. Discrepancy between studies could be related to the use of different sequencing library kits, indeed in the previous comparison, the Rapid sequencing kit (SQK-RAD004) was used for ONT library preparation, while in this comparison, we used the Ligation sequencing kit (LSK109). The LSK109 kit includes a DNA repair step using NEB's NEBNext FFPE DNA Repair Mix, which potentially corrects for sequence differences observed with the Rapid kit. Although the Rapid sequencing kit reduces handling time, the ligation sequencing approach appears to offer a lower final error rate.

To maximize sequencing costs and processing time, it is of paramount importance to evaluate the minimum read depth required to assemble an accurate OsHV-1 genome. Using the approach tested here, we found that coverage of 50 OsHV-1 ONT long reads was sufficient to obtain a complete OsHV-1 genome with more than 99.99% accuracy compared to the Illumina genome. In other words, only 6,061 OsHV-1 long reads with an N50 of 3 kb were required to assemble a complete and accurate OsHV-1 genome. Assuming that 5% of the data sequenced from infected tissues is from OsHV-1, it would take approximately 3 hours to generate an accurate OsHV-1 genome.

An alternative approach to rapidly obtain accurate genomes is to bypass time-consuming *de novo* assembly and polishing steps by directly calling consensus sequences after aligning reads to a reference genome. In our study, we assessed the performance of five consensus callers (NanoCaller, Clair3, Longshot, Medaka, and Pepper DeepVariant) in calling an accurate OsHV-1 genome using ONT reads from different DNA extraction kits and sequencing runs.

Since the same viral inoculum was used for all ONT samples, and both the Illumina and ONT assembled genomes showed minimal differences (15 bp), the observed percentage of identity between the consensus sequences and the Illumina OsHV-1 genome primarily reflects ONT sequencing errors and the ability of consensus callers to generate accurate consensuses. Notably, we found that the accuracy of certain consensus callers was directly influenced by read depth.

NanoCaller demonstrated higher accuracy with low read depth samples, enabling the generation of accurate consensus sequences from limited data, which could be advantageous for real-time genome calling of individuals with low levels of viral infection. Conversely, Longshot performed exceptionally well in high-read depth scenarios, generating accurate consensus sequences from extensive data, particularly in the later stages of infection. Clair3, Pepper DeepVariant and, to a lesser extent, Medaka, showed consistent performance, generating accurate consensus sequences from all samples regardless of sequencing depth. These results are consistent with previous studies that have observed similar results regarding read depth and tool performance (34–37).

Pairwise comparisons between the Illumina and ONT genomes, along with results from minimum sequencing depth and consensus calling, strongly suggest that ONT is an efficient method for assembling a complete and accurate OsHV-1 genome. This is achieved without the need for a hybrid approach (i.e., without the need for both long- and short-read sequencings) and with minimal bias from sequencing errors.

In addition to its efficiency at the base level, ONT sequencing allowed us to detect structural variations within the OsHV-1 genome and to characterize and quantify the different isomers. While four major OsHV-1 isomers are characterized by inversions of the $U_L$ and $U_S$ regions, three minor isomers are characterized by the absence of the X region or its displacement at the 5′ end of the $TR_L$ region and between the $IR_L$ and $IR_S$ regions (1). Recently, we have reconfirmed the presence of the four major isomers in equimolar proportions within viral particles (18). Here, similar results were obtained from infected tissues. In addition, a few reads were also missing in the X region in this new data set. However, this observation is more likely due to a higher sequencing depth rather than phenotypic differences between genomes sequenced from infected tissues and viral particles. This is the first time such an observation has been made from viral DNA isolated from infected tissues since the original OsHV-1 genome was sequenced (1). To better quantify and characterize the major isomers, it will be essential to obtain higher quantity of reads long enough to span the $U_L$-$IR_L$-X-$IR_S$-$U_S$ regions. This could be achieved by adding an additional step prior to sequencing library preparation. For example, a short fragment elimination step was applied in run 4, resulting in a higher proportion of longer reads. However, these reads were still not long enough to capture all the regions affected by isomerization. Alternative strategies to enrich for longer DNA fragments prior to sequencing include gel purification after electrophoresis or the use of automated size selection systems such as BluePippin coupled with the ONT Ultra-Long DNA sequencing kit. Although effective, these methods will not reduce the overall sequencing effort or cost. SPRIselect Bead for short fragment elimination could be an alternative cost-effective solution, but the loss of total DNA during magnetic bead-based purification implies a large amount of input DNA. For diagnostic purposes and rapid genomic characterization, we suggest that HMW DNA extraction with a dedicated commercial kit is probably sufficient for a good OsHV-1 genomic characterization.

In this study, none of the isomer identification methods used were able to confirm the presence of all minor isomers in the OsHV-1 read population because none of the reads included the entire $U_L$ region, making it impossible to determine whether the X region located at the 5′ end of the $TR_L$ region is a true minor isomer or just a result of the inverted $U_L$ isomer.

The study of structural variation among herpesviruses is critical to understanding their complete viral life cycles. While OsHV-1 and HSV-2 show equimolar fractions of major isomers in both viral particles and infected tissues (38), HSV-1 shows variable isomer proportions depending on the cell type and strain infected (39), whereas VZV shows differences in the proportions of inverted short- and long-region isomers (40). Understanding these structural variations is essential for the comprehensive characterization of herpesvirus life cycles.

Finally, in a previous study, we showed that TFF enabled to co-purification of several DNA viruses in addition to OsHV-1 (18). By contrast, relatively few DNA viruses were identified in this study. This is likely because the analysis was performed on infected tissues which resulted in a high proportion of host DNA being sequenced, reducing the detection of other DNA viruses. In addition, the use of AS to enrich OsHV-1 DNA further limited the sequencing of other organisms or viruses.

The use of ONT sequencing coupled with AS allowed us to sequence OsHV-1 genomes directly from host tissues. Thanks to the availability of an OsHV-1 reference genome, this approach allowed enrichment of viral genomes without the constraints of laborious laboratory steps required to get enough viral particles isolated from the host tissues.

This advanced sequencing technology is a promising solution to rapidly and cost-effectively detect OsHV-1 in infected oyster tissues, to better characterize its life cycle, and to provide an efficient diagnostic tool to detect unculturable aquatic viruses directly from infected host tissues. However, the technology and bioinformatics tools used in this study continue to evolve. Recent developments include the release of updated flow cells and advances in *de novo* assembly and consensus calling algorithms. Consequently, the results presented here could potentially be improved by using the latest versions of flow cells and bioinformatics tools.

## MATERIALS AND METHODS

### Short-read Illumina sequencing

#### *OsHV-1 natural infection*

During a mortality event in August 2020, highly susceptible standardized oyster spat were deployed in the field at the "La Floride" site in the Marennes-Oléron Bay, France (Lat 45.80° and Long −1.15°). Once mortality was detected, the spat was brought to the laboratory, where they were monitored twice daily to collect fresh moribund oysters.

#### *DNA extraction and library preparation*

About 35 mg of the mantle was collected from one moribund oyster. Nucleic acid extraction was performed using the Magattract HMW kit (Qiagen) according to the manufacturer's protocol. DNA purity and concentration were measured using the Nanodrop ND-1000 spectrometer (Thermo Scientific) and Qubit dsDNA BR assay kits (Molecular Probes Life Technologies). Before sequencing, DNA was tested for viral copy number by qPCR (see below). Illumina DNA-Seq library preparation and sequencing were performed by Genome Quebec Company (Genome Quebec Innovation Center, McGill University, Montreal, Canada) as previously described (18), using 120 ng of DNA extracted from the infected oyster, using the Shotgun PCR-Free Library Kit (Lucigen) and the NovaSeq 6000 Sequencing device (Illumina) (paired ends, 150 bp) (Fig. 12).

### Long-read ONT sequencing

#### *Oyster production*

Highly susceptible *M. gigas* spat were produced at the Ifremer hatchery in Bouin in June 2020 (Vendée, France) as described in reference (41). Oysters were always protected in our facilities using filtered and UV-treated seawater, and so they never experienced mortality outbreaks. Prior to the experiment, spat (9 months old) were acclimatized at 19°C for 2 weeks in flow-through 120 L tanks using filtered and UV-treated seawater

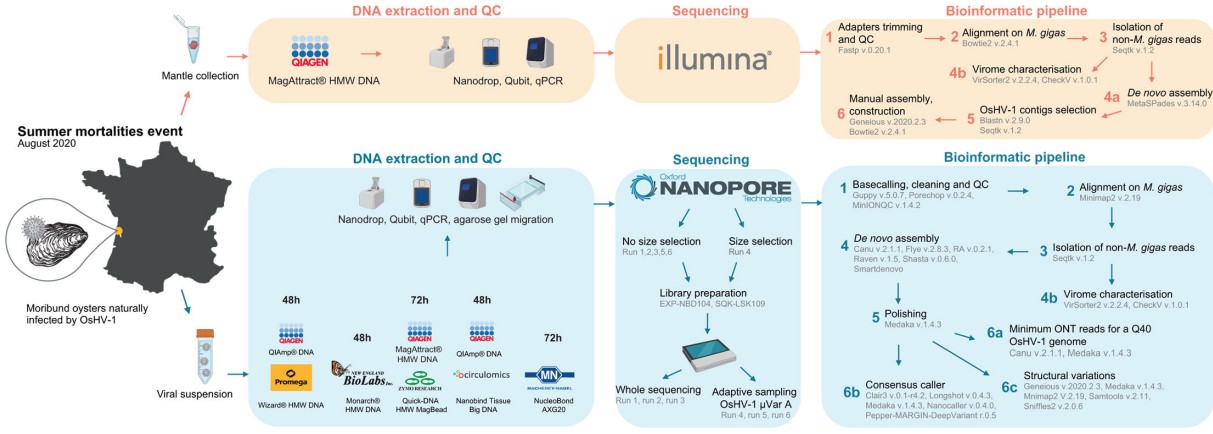

**FIG 12** Experimental design and bioinformatic pipeline implemented in this study.

enriched in phytoplankton (*Skeletonema costatum, Isochrisis galbana*, and *Tetraselmis suecica*).

## Viral inoculum and experimental OsHV-1 infection

From the same cohort as the one sampled for the Illumina sequencing (see above), the OsHV-1 strain used for the experimental infection was primarily isolated from moribund-infected oysters collected on 5th August 2020 and was then successively experimentally inoculated twice in juvenile oysters. The virus was isolated from the gills and mantle of moribund oysters infected with OsHV-1 on 25th January 2021. Viral inoculum was prepared following previously described protocols (31) and, for each oyster, 100 µL aliquot of OsHV-1 inoculum ($9.3.10^5$ OsHV-1 DNA copies $µL^{-1}$) was injected into the adductor muscle. In total, 50 juvenile oysters were used.

## Sampling and tissue dissection

Oysters were checked 48 hours and 72 hours post-OsHV-1 injection, and each moribund oyster (47 in total) was dissected for gills and mantle isolation. Dissections were then subjected to manual homogenization with plastic pestles. Homogenates were then carefully weighed and stored on ice until DNA extraction.

## DNA extraction

For each sample, between 18 mg and 25 mg of homogenized tissues was subjected to DNA extraction with one or two of the seven commercial DNA extraction kits. Six of these kits were specifically designed for the extraction of high molecular weight (HMW) DNA: the Promega Wizard HMW DNA Extraction kit (hereafter referred to as Promega, P) which uses centrifugation for the DNA cleaning step; the New England Biolabs Monarch HMW DNA Extraction kit for Tissue (hereafter referred as Monarch, M), which uses glass beads to retain DNA during the cleaning steps; the Zymo Research Quick-DNA HMW MagBead kit (hereafter referred as Zymo, Z) and the QiAgen MagAttrac HMW DNA kit (hereafter referred as MagAttract, Ma), both of which use magnetic beads for the DNA cleaning steps; the Circulomics Nanobind Tissue Big DNA kit (hereafter referred as Circulomics, C), which uses magnetic disk to retain DNA during the cleaning steps; and the Macherey Nagel NucleoBond AXG 20 kit (hereafter referred as Macherey Nagel, Mn) which uses a gravity column to extract DNA. The last kit used was the QiAgen QIAmp DNA kit (hereafter referred to as QiAgen, Q, and Q'), which is designed for the recovery of regular-weight DNA and uses silica columns for DNA purification.

Each extraction kit was tested using 5–7 moribund oysters (Table 2).

Each DNA extraction procedure was carried out according to the manufacturer's protocol. To preserve HMW DNA as much as possible, the main modifications involved replacing pipette/vortex mixing with gentle flicking. In addition, to limit mechanical stress on DNA during the lysis and RNAse treatment steps of the Circulomics and Monarch kits, we reduced the agitation speed to 250 rpm (from 900 and 300 rpm, respectively). Following DNA extraction with the different kits, all purified DNA was stored at 4°C until further processing, avoiding unnecessary freeze-thaw cycles.

**TABLE 2** Sample description

| Biological material | OsHV-1 infection | Number of oysters sampled | DNA extraction kit | Sequencing technology |
|---|---|---|---|---|
| Highly susceptible standardized oyster spat | Natural | 1 | MagAttract | Illumina |
| Naive oysters | Injection | 5 | Promega (P) and QiAgen (Q) | ONT |
| Naive oysters | Injection | 7 | Monarch (M) | ONT |
| Naive oysters | Injection | 7 | Zymo (Z) | ONT |
| Naive oysters | Injection | 7 | MagAttract (Ma) | ONT |
| Naive oysters | Injection | 6 | Circulomics (C) and QiAgen (Q') | ONT |
| Naive oysters | Injection | 6 | Macherey Nagel (Mn) | ONT |

## DNA quality control

The purity of freshly extracted DNA was evaluated by a Nanodrop spectrophotometer, and the DNA concentrations were measured using the Qubit dsDNA BR Assay kit. The distribution size of the DNA fragments was then evaluated by electrophoresis with the migration of 300 ng of purified DNA on a 0.6% agarose gel containing ethidium bromide for 2.5 hours at 112 V. The agarose gels were also loaded with Quick-Load 1 kb Extend DNA Ladder, which has sizes ranging from 0.5 to 48.5 kpb. These agarose gels were then visualized under UV light and the images were captured with the ImageLab 3.0 software.

## OsHV-1 genome copy number estimation by qPCR

Quantification of the OsHV-1 genome copy number was performed using quantitative PCR (qPCR) using the primers OsHVDP For (forward) 5′-ATTGATGATGTGGATAATCTGTG-3′ and OsHVDP Rev (reverse) 5′-GGTAAATACCATTGGTCTTGTTCC-3′ (42) (18). The absolute quantification of viral genome copy number was estimated by comparing the observed Ct value with a standard curve obtained using a serial dilution of plasmid DNA (pUC57) including the target region (equivalent to $1.10^5$ to $1.10^1$ OsHV-1 DNA copies $\mu L^{-1}$). Each qPCR run included a complete standard curve and a non-template control (NTC). All reactions were performed with three technical replicates. Data were then collected on the MxPro software, and the dissociation curves were verified. Reaction efficiency was confirmed to be between 90% and 110% with $R^2$ >90. Finally, technical replicates with Ct differences over 0.5 were rejected. The OsHV-1 genome copy number was then calculated for all DNA-extracted samples for each extraction kit.

## Library preparation and ONT sequencing

ONT sequencing libraries were prepared using the Ligation Sequencing Kit (SQK-LSK109) coupled with the PCR-free Native Barcoding Expansion Kit (EXP-NBD104 and EXP-NBD114) for multiplexing following the ONT protocol "Native barcoding genomic DNA" (Version: NBE_9065_v109_revAA_14Aug2019). The libraries were loaded on MinION flow cells (FLO-MIN106D, R9.4) and sequenced on a MinION Mk1C device using the MinKNOW software version 21.02.2.

To determine the ability of ONT to sequence OsHV-1 from infected oyster tissues, we performed a total of six ONT sequencing runs, each with seven multiplexed samples corresponding to DNA extracted with the seven different extraction kits. Three ONT sequencing runs were performed on the whole DNA samples (hereafter named "whole sequencing") and three ONT sequencing runs were performed with AS option activated with enrichment on the OsHV-1 µVarA reference genome (GenBank: KY242785.1) to enrich for OsHV-1 DNA molecules in real time during the sequencing process.

Flow cells used for two different runs were washed using the Flow Cell Wash kit (EXP-WSH004), allowing them to be re-used in subsequent runs. In addition, libraries loaded consecutively on the same washed flow cell were built with different barcode sets. Each sequencing run was set up as described in Table 3. Run 1 was performed on samples with the best ratio of the OsHV-1 genome copies per ng of extracted DNA while the five other runs were performed with the best apparent migration profile on agarose. In addition, a short fragment elimination step, with the Circulomics Short Read Eliminator kit (10 kb), was performed for run 4 ( Table 3; Fig. 2).

**TABLE 3** Description of the six sequencing runs

|  | Duration (h) | Flowcell | Available pores | Samples | Adaptive sampling |
|---|---|---|---|---|---|
| Run 1 | 48 | Used | 990 | P1, M5, Q'1, Z1, Ma2, C1, and Mn1 | No |
| Run 2 | 39 | New | 1488 | P3, M5, Q'5, Z4, Ma7, C6, and Mn5 | No |
| Run 3 | 26 | Used | 189 | P2, M4, Q3, Z7, Ma5, C4, and Mn3 | No |
| Run 4 | 16 | New | 1,352 | P3, M5, Q3, pool of Z1 and Z2, Ma6, C6, and Mn6 | Yes |
| Run 5 | 40 | Used | 341 | P2, M4, Q3, Z7, Ma5, C4, and Mn3 | Yes |
| Run 6 | 4 | Used | 1,054 | P2, M4, Q3, Z7, Ma5, C4, and Mn3 | Yes |

After sequencing, all fast5 data generated by the Mk1C with the MinKNOW software were transferred to the Ifremer HPC cluster Datarmor for data analysis (Fig. 12).

## Data analysis

### *Illumina assembly*

Illumina OsHV-1 genome assembly has been performed using a previously developed bioinformatic pipeline (18). This pipeline is summarized in Fig. 12.

### *ONT basecalling, cleaning, quality control, and reads sorting*

The resulting fast5 reads were basecalled with Guppy GPU version 5.0.7 with the following parameters: --config dna_r9.4.1_450bps_hac.cfg --detect_mid_strand_barcodes, --min_score_mid_barcodes 50, --min_score 50, --trim_threshold 50, -- trim_barcodes and --barcode_kits "EXP-NBD104 EXP-NBD114." The fastq reads obtained after basecalling were cleaned with Porechop v0.2.4 (https://github.com/rrwick/Porechop) to remove the remaining adapters and barcodes. To this end, the adapter.py file containing reference adapters and barcode sequences used by Porechop was modified to include the sequences of the adapters from the SQK-LSK109 kit. Porechop was run with the following parameters: --adapter_threshold 78, --middle_threshold 78 and --extra_middle_trim_bad_side 10 --check_reads 100000. The resulting clean fastq reads were then subject to quality control analysis with MinIONQC v1.4.2 (43). To determine the origins of the read (*M. gigas* versus OsHV-1), we performed a read classification by mapping the ONT reads on the *M. gigas* reference genome (GenBank: GCA_902806645.1) and on the OsHV-1 µVar A reference genome (GenBank: KY242785.1) using Minimap2 v2.19 (44) with the parameter: -x map-ont. The alignment files were then compressed, sorted, and indexed using Samtools v1.11 (45). Custom homemade scripts were then used to extract non-*M. gigas* reads, which were used for OsHV-1 *de novo* assembly (Fig. 12).

### *ONT OsHV-1 de novo assembly*

Six long-read *de novo* assemblers were tested: Canu version 2.1.1 (46) with standard parameters; Flye version 2.8.3 16 (47) with the parameter --meta; Shasta version 0.6.0 (https://github.com/paoloshasta/shasta) with standard parameters; Smartdenovo (https://github.com/ruanjue/smartdenovo) with parameter: -J 200; Raven version 1.5 (https://github.com/lbcb-sci/raven) with standard parameters and RA version 0.2.1 (https://github.com/lbcb-sci/ra) with standard parameters. Each assembler was run: (i) on a pool of reads corresponding to all the non-*M. gigas* reads from all the tested kits and all the sequenced runs, (ii) on the top 10% longest reads, (iii) on the top 1% longest reads, and (iv) on the top 0.1% longest reads. OsHV-1 *de novo* assembled contigs were then identified with BlastN (48) against the OsHV-1 µVar A genome (GenBank: KY242785.1) and curated in Geneious version 2020.2.4 (https://www.geneious.com). The curated OsHV-1 genome was then consensus called with all non-*M. gigas* reads using Medaka version 1.4.3 with parameters medaka_consensus and –m r103_sup_g507 (https://github.com/nanoporetech/medaka) (Fig. 12).

To evaluate the minimum ONT reads needed to obtain a complete and accurate OsHV-1 genome, we generated several fastq data sets, containing non-*M. gigas* reads from all runs, each corresponding to a targeted read depth on the OsHV-1 genome. We used a combination of Canu *de novo* assembly version 2.1.1 and Medaka consensus calling version 1.4.3 (https://github.com/nanoporetech/medaka) on (i) 99 fastq data sets with read depth ranging from 20× to 1,000× and (ii) 56 fastq data sets with read depth ranging from 1,100× to 6,700×. Each assembly was separately compared to the OsHV-1 genome assembled from Illumina to evaluate its quality through the ANI between the two genomes.

## Virome characterization

For Illumina data, *de novo* assembly was performed on non-OsHV-1 and non-oyster reads using SPades v.4.0.0 (49) with --meta parameter. Contigs longer than 1 kb were then collected using SeqTK v.1.4 (https://github.com/lh3/seqtk). Viral sequence identification was performed on *de novo* assembled contigs using VirSorter2 v.2.2.4 (50), CheckV v.1.0.1 (51), and DRAM v.1.5.0 (52), as described in Sullivan Lab protocol (dx.doi.org/10.17504/protocols.io.bwm5pc86). Contigs containing at least one viral gene were classified as viral contigs. Contigs that did not contain a viral gene but met one of the following criteria were also retained: (i) no viral gene and no host gene, (ii) no viral gene and a score greater than 0.95, or (iii) no viral gene and a hallmark gene counts greater than 2 (i.e., highly conserved and characteristic of a particular virus or group of viruses). "Suspicious" genes were further checked with DRAM, as described in the protocol.

For ONT data, *de novo* assembly was performed on non-OsHV-1 and non-oyster reads using Flye v.2.9 (47) with --nano-raw and --meta parameters. Medaka v.1.4.3 (https://github.com/nanoporetech/medaka) was then used to polish all contigs. Contigs longer than 1 kb were then collected using SeqTK v.1.4 (https://github.com/lh3/seqtk). Viral sequence identification was performed on *de novo* assembled contigs using the same approach applied to Illumina contigs.

## Comparison of Illumina and ONT OsHV-1 genome sequencing

Before whole-genome alignment, both genomes were modified to fit the same isomer version (i.e., P isomer). To evaluate the accuracy of ONT sequencing, the manually constructed Illumina OsHV-1 Non-Redundant (NR) genome was aligned with the manually curated ONT OsHV-1 complete genome using Mafft version 1.4.0 with default parameters (53) implemented in Geneious version 2020.2.4 (https://www.geneious.com).

## Evaluation of long-read consensus callers

For consensus caller evaluation, reads from each sample, each kit, and each run were mapped (i) individually (42 alignments); (ii) by run (sum of seven samples: six alignments); and (iii) all sequenced reads (sum of the six runs: one alignment) on the new *de novo* assembled and consensus called ONT OsHV-1 genome. A total of 49 alignment files were consensus called with five long-read consensus callers: Pepper-MARGIN-DeepVariant version r0.5 (36) with parameter: --ont; Nanocaller version 0.4.0 (34) with parameter: -seq ont; Medaka version 1.4.3 (https://github.com/nanoporetech/medaka) with parameter: medaka_haploid_variant; Longshot version 0.4.3 (54) with parameter --no_haps; and Clair3 version 0.1 (55) with parameters: --chunk_size = 205000, --no_phasing_for_fa, --include_all_ctgs and --haploid_precise. For each consensus caller, 49 consensus sequences were then produced using Bcftools version 1.10.2 (45) with standard parameters of bcftools consensus. Mafft version 7.313 with standard parameters was used to create an alignment of the 49 consensus sequences along with five other OsHV-1 genome sequences (i) the OsHV-1 genome from 2005 (GenBank: NC005881.2), (ii) the OsHV-1 μVar A sequence (GenBank: KY242785.1), (iii) the OsHV-1 genome assembled with Illumina reads and both (iv) non-polished, and (v) polished OsHV-1 genomes assembled with ONT reads. Five pairwise percentage identity matrices (one for each consensus caller) of the 54 OsHV-1 sequences were then extracted with Geneious v2020.2.4 (https://www.geneious.com) (Fig. 12).

## OsHV-1 structural variation and isomers quantification

For structural variation detection and OsHV-1 isomer characterization, we evaluated two methods. First, to quantify OsHV-1 reads corresponding to different isomers, we used the Canu *de novo* assembly using the top 0,1% of the longest reads, which produced four contigs each representing one of the four major isomers (P: $U_L \rightarrow$/ $U_S \rightarrow$; $I_S$:$U_L \rightarrow$/ $U_S \leftarrow$; $I_{LS}$: $U_L \leftarrow$/ $U_S \leftarrow$ and $I_L$: $U_L \leftarrow$/ $U_S \rightarrow$). They were trimmed on the $U_L$ side to retain a 10 kb $U_L$ fragment with Geneious version 2020.2.4 (https://www.geneious.com). The

four major isomers were then polished with Medaka version 1.4.3 (https://github.com/nanoporetech/medaka) using all the non-*M. gigas* reads and were used as a reference for mapping all reads aligning on the ONT assembled genome using Minimap2 version 2.19 with parameters -ax map-ont, --sam-hit-only, and --secondary=no. Samtools version 2.11 was then used to filter out all the shortest aligned reads and all alignments with MapQ <40, retaining only the reads spanning at least a fragment of the $U_L$ and a fragment of the $U_S$ (i.e., 28 kb long). Second, to evaluate the detection of structural variations (SV) within the OsHV-1 genome, we used Sniffles2 version 2.0.6 (56) with the parameters: --minsupport 1 and --no-qc, to detect all SV regardless of the number of reads supporting the SV or the quality of the SV. We used two types of alignments as input for Sniffles 2: (i) all ONT reads of all runs on the polished ONT OsHV-1 genome and (ii) ONT reads from each DNA extraction kit of each run on the polished ONT OsHV-1 genome. We then filtered the SVs, keeping only those that were flagged as "PRECISE," longer than 500 bp, and supported by at least two reads (Fig. 12).

## Visualization

We generated all the figures using R (https://www.r-project.org) with ggplot2 (https://ggplot2.tidyverse.org) or using Geneious. We prepared the figures using Inkscape (https://inkscape.org/fr).

## ACKNOWLEDGMENTS

We thank the staff of the Ifremer station at La Tremblade (ASIM), Frédéric Girardin and his team (Plateforme des Mollusques Marins de La Tremblade PMMLT), and Christophe Stavrakakis and his team (Plateforme des Mollusques Marins de Bouin PMMB). We also thank the SEBIMER team for maintaining bioinformatics tools.

C.P. was financially supported by a grant from the Ifremer Scientific Board and the Nouvelle-Aquitaine region. A.D.M. was financially supported by Ifremer, EU DG SANTE through the European Reference Laboratory for Mollusc Diseases, and DGAL through the National Reference Laboratory for Mollusc Diseases. The authors acknowledge the Pôle de Calcul et de Données Marines (PCDM; https://wwz.ifremer.fr/en/Research-Technology/Research-Infrastructures/Digital-infrastructures/Computation-Centre) for providing DATARMOR computing and storage resources. The funders had no role in study design, data collection and interpretation, or the decision to submit the work for publication.

G.C., B.M., C.P., S.H., and A.D.M. designed the study; L.D., B.M., and C.P. provided the biological materials; A.D.M., G.C., and S.H. processed the samples. A.D.M. and C.G. performed ONT sequencing. C.P. and G.C. performed the bioinformatic analysis of the Illumina data, and A.D.M. and G.C. performed the bioinformatic analysis of the ONT data. A.D.M., G.C., and C.P. drafted the manuscript and B.M. and I.A. corrected it. All authors read and approved the final version of the manuscript.

## AUTHOR AFFILIATION

[1]Ifremer, ASIM, La Tremblade, France

## AUTHOR ORCIDs

Aurélie Dotto-Maurel  http://orcid.org/0000-0002-2930-3078
Germain Chevignon  http://orcid.org/0000-0002-8880-8615

## AUTHOR CONTRIBUTIONS

Aurélie Dotto-Maurel, Conceptualization, Data curation, Methodology, Software, Visualization, Writing – original draft, Writing – review and editing | Camille Pelletier, Conceptualization, Data curation, Formal analysis, Writing – original draft, Writing – review and editing | Lionel Degremont, Resources, Writing – review and editing | Serge Heurtebise, Methodology, Writing – review and editing | Isabelle Arzul, Writing

– review and editing | Benjamin Morga, Conceptualization, Resources, Validation, Writing – original draft, Writing – review and editing | Germain Chevignon, Conceptualization, Data curation, Formal analysis, Software, Validation, Visualization, Writing – original draft, Writing – review and editing

## DATA AVAILABILITY

All sequence data have been deposited in ENA. ONT data are accessible on ENA under the Bioproject accession number PRJEB82928 via https://doi.org/10.12770/1a301465-aee7-42c2-a950-7708f422a6aa (Run1: SAMEA117397672, Run2: SAMEA117397675, Run3: SAMEA117397676, Run4: SAMEA117397673, Run5: SAMEA117397674, and Run6: SAMEA117397676) and Illumina data are accessible on ENA under the Bioproject accession number PRJEB81586 via https://doi.org/10.12770/e12a5d05-cc95-407f-b0f7-c86e79cc5b3a and sample accession: SAMEA117373467.

The ONT assembly genome is available at GenBank accession GCA_964656485.1, while the Illumina assembly genome is available at GenBank accession GCA_964656475.1.

Analysis scripts are publicly available at https://gitlab.ifremer.fr/lgpmm/ont-bioinfor-matic-tools-for-oshv-1-genome-characterization.

## ADDITIONAL FILES

The following material is available online.

### Supplemental Material

**Supplemental figures and tables (Spectrum02082-24-s0001.pdf).** Fig. S1, S2, and S3; Table S1.

### Open Peer Review

**PEER REVIEW HISTORY (review-history.pdf).** An accounting of the reviewer comments and feedback.

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
