## [Reviewer comments · Microbiology Spectrum]

Microbiology Spectrum

Evaluation of long-read sequencing for Ostreid herpesvirus type 1 genome characterization from *Magallana gigas* infected tissues

Aurélie Dotto-Maurel, Camille Pelletier, Lionel Dégremont, Serge Heurtebise, Isabelle Arzul, Benjamin MORGA, and Germain Chevignon

Corresponding Author(s): Germain Chevignon, Ifremer

Review Timeline:

Submission Date:	August 21, 2024
Editorial Decision:	October 13, 2024
Revision Received:	December 15, 2024
Accepted:	December 19, 2024

Editor: Zsolt Toth

Reviewer(s): Disclosure of reviewer identity is with reference to reviewer comments included in decision letter(s). The following individuals involved in review of your submission have agreed to reveal their identity: Zsolt Csabai (Reviewer #1); Guillaume Croville (Reviewer #2); Umberto Rosani (Reviewer #3)

Transaction Report:

DOI: <https://doi.org/10.1128/spectrum.02082-24>

Re: Spectrum02082-24 (Evaluation of long-read sequencing for Ostreid herpesvirus type 1 genome characterization from Magallana gigas infected tissues)

Dear Dr. Germain Chevignon:

Thank you for the privilege of reviewing your work. Your manuscript was evaluated by three independent peer reviewers. Below you will find my comments, instructions from the Spectrum editorial office, and the reviewer comments.

The reviewers found the study important but identified some aspects of the manuscript that should be improved. We therefore ask you to modify the manuscript according to the review recommendations before we can consider your manuscript for acceptance. Your revisions should address the specific points made by each reviewer.

Revision Guidelines

Sincerely,
Zsolt Toth
Editor
Microbiology Spectrum

Reviewer #1 (Comments for the Author):

The authors have isolated DNA from OshV-1 virus-infected oysters using different isolation kits and, after evaluating the

efficiency of these kits, sequenced and assembled the genome using the Oxford Nanopore Ligation Sequencing Kit and various freely available programs. The comparison of the programs used also presents interesting differences that could be useful for those facing similar challenges, particularly with the difficulty of sequencing terminal repeats using short-read sequencing, which necessitates the use of long-read sequencing methods.

The publication describes a well-designed, logical experimental approach, utilizing seven different commercially available DNA isolation kits. What stands out more is the comparison of programs that handle long reads, clearly demonstrating the advantages and disadvantages of various freely available tools when dealing with viruses.

One critical observation regarding the publication is that the experiment, as described, is not reproducible. The sequencing data were likely generated several years ago, as the ONT LSK109 ligation kit was used for library preparation. Since then, both the LSK110 and LSK112 kits have been discontinued, and only the LSK114 kit, which features the improved q20+ Chemistry, is available, providing much better read quality during sequencing and base-calling. The same applies to the R9 flow cell type, though it was only phased out of production this year, so I won't consider that a major issue.

Another concern is that the software used for the analysis is now available in much more recent versions:

Shasta: version 0.6 has been updated to 0.13.0

Raven: version 1.5 is now at 1.8.3

Canu: version 2.1.1 has been updated to 2.2

Q1. My first question relates to this: do the newer versions of these genome assembly programs affect the results obtained? (It's also worth noting that Medaka has released an update just a few weeks ago, so that might be relevant too.)

Q2. Regarding Figure Supplementary 1: Does the "Available pores" refer to the result of the flow cell check, or does it represent the number of pores that were actively sequencing after the library was loaded? This question arises because there seems to be no correlation between the initial pore count and the amount of bases obtained.

I also have a few structural remarks about the publication:

Line 24: HWM is written instead of HMW in the abbreviation.

Line 62: There is an unnecessary parenthesis after the word "Us."

Lines 103/104: The sentence here is duplicated.

Line 499: SQS is written instead of SQK-LSK109.

Reviewer #2 (Comments for the Author):

The authors present a study aimed at improving the sequencing performance of the OsHV-1 virus, responsible for considerable economic losses in the Magallana gigas Pacific oyster industry. It is therefore necessary to produce genomic data enabling the biology of the virus to be studied in detail.

The authors' work is justified by the problems and difficulties of sequencing viruses with large DNA genomes containing repeats and inversions. Shotgun long reads metagenomics is therefore the method of choice for avoiding the need for complex techniques such as TFF, especially when dealing with a non-cultivable virus.

Comparisons of the various kits and bioinformatics algorithms are extremely rigorous, both in methodology and in their description in this article. The evaluation of adaptive sampling is very interesting and promising, which is reassuring because this novelty is not always robust depending on the desired application.

Minor corrections:

Line 24: HMW instead of HWM.

Lines 104-105: sentence repeated.

Lines 389-390: please rephrase.

Lines 440-441: Explain why you reduce agitation to 250 rpm and whether you are basing this on experiments.

Line 506: Can you describe the homemade scripts in detail?

Lines 526-527: If you're talking about sequencing depth, it would be preferable to use this term instead of "coverage", using the "X" as the unit.

Given that the authors have shotgun metagenomic data in their possession, it would be interesting to have results exploiting this data by providing the DNA virome of the samples.

Finally, in view of their colleagues' publication establishing a link between telomerase activity and OsHV-1 infection (Dupoué et al. 2024), it would be interesting for the authors to discuss the potential of their developments in this context.

Reviewer #3 (Comments for the Author):

Dotto-Maurel and colleagues proposed the paper titled "Evaluation of long-read sequencing for Ostreid herpesvirus type 1 genome characterization from Magallana gigas infected tissues". This study is a comparison of different DNA extraction methods

and sequencing technologies to reconstruct the OsHV-1 genome. The analysis is interesting and the experimental approach is appropriate, with an adequate number of tested methods. The use of different assemblers to reconstruct the viral genome is also of interest.

The main focus of the study was to develop and test a methodology to effectively enrich for viral DNA over host molecules, in order to propose it as a future diagnostic method for aquaculture. To this end, the use of ONT sequencing over other long-read technologies (e.g. PacBio) is fully justified. However, I guess that briefly mentioning the existence of this other technologies in the introduction might be beneficial, as well as reporting if long-read sequencing has been previously used to investigate OsHV-1 pathogenesis.

I have a single major concern, whereas several minor points are reported below.

The author evaluated the DNA fragmentation running DNA extract on an agarose gel, with a marker with a maximal size of 48.5 kb. Several extraction kits produced HMW DNA located at the very beginning of the run. I guess that using a TapeStation (Agilent) instrument will probably generate more useful results, with the possibility to evaluate also higher sizes (e.g. up to 60 kb), which might differentiate the results.

Moreover, since the averaged length of the ONT reads is shorted that the distribution of DNA extracts (at least as it can be evaluated by the gel picture), the authors should comment this aspect. Can other approaches used to further enrich for longer DNA fragments before sequencing, thus reducing sequencing efforts and costs?

Minor points:

Line 47. The name of the family should be italicized

Line 77-78. ONT is not the only technology able to produce long reads.

Line 92. Perhaps add "electric"

Line 96-99. This specification seems trivial and more suitable for a protocol than for an introduction.

Line 139. What is the definition of "best ratio"?

Line 165. The origin of the DNA used for Illumina sequencing is, at this point, not defined. Maybe it should be briefly introduced.

Line 275. Correct 13,x into 13.x

Methods. Since the flow cells were washed and reused again, it would be interesting to have somewhere reported the number of pores available at the start of each sequencing run and if their decline influence the sequencing readout.

Figure 4. What is the origin of the reads classified as "OsHV-1+oyster"?

Figure 5. Although this figure contains valuable information, the comparison between conditions is really hard (using a 1-page magnification). Perhaps a different visualization could be consider the allow a more immediate understanding of the differences.

Evaluation of long-read sequencing for Ostreid herpesvirus type 1 genome characterization from *Magallana gigas* infected tissues

Reviewer #1 (Comments for the Author):

The authors have isolated DNA from OsHV-1 virus-infected oysters using different isolation kits and, after evaluating the efficiency of these kits, sequenced and assembled the genome using the Oxford Nanopore Ligation Sequencing Kit and various freely available programs. The comparison of the programs used also presents interesting differences that could be useful for those facing similar challenges, particularly with the difficulty of sequencing terminal repeats using short-read sequencing, which necessitates the use of long-read sequencing methods.

The publication describes a well-designed, logical experimental approach, utilizing seven different commercially available DNA isolation kits. What stands out more is the comparison of programs that handle long reads, clearly demonstrating the advantages and disadvantages of various freely available tools when dealing with viruses.

One critical observation regarding the publication is that the experiment, as described, is not reproducible. The sequencing data were likely generated several years ago, as the ONT LSK109 ligation kit was used for library preparation. Since then, both the LSK110 and LSK112 kits have been discontinued, and only the LSK114 kit, which features the improved q20+ Chemistry, is available, providing much better read quality during sequencing and base-calling. The same applies to the R9 flow cell type, though it was only phased out of production this year, so I won't consider that a major issue.

Another concern is that the software used for the analysis is now available in much more recent versions:

Shasta: version 0.6 has been updated to 0.13.0

Raven: version 1.5 is now at 1.8.3

Canu: version 2.1.1 has been updated to 2.2

Q1. My first question relates to this: do the newer versions of these genome assembly programs affect the results obtained? (It's also worth noting that Medaka has released an update just a few weeks ago, so that might be relevant too.)

R1: *Although newer versions of the software and kits have been released, the versions we used (kit LSK109, R9 flow cells, Shasta 0.6, Raven 1.5, Canu 2.1.1 and also for all other software used in this study) provided reliable and reproducible results. Newer versions of software or chemistry don't invalidate previous results; they simply offer improvements for future studies. Re-analysis of all data using the latest kits and software versions would require considerable resources and the potential impact on the final results is likely to be minimal. The main objective of the study was to validate the use of ONT sequencing to assemble and characterise the OsHV-1 genome, and it is unlikely that the use of newer software versions would fundamentally alter the biological conclusions. The existing results are strong and would remain valid regardless of recent improvements in sequencing technologies. This has been added in lines 421-427.*

Q2. Regarding Figure Supplementary 1: Does the "Available pores" refer to the result of the flow cell check, or does it represent the number of pores that were actively sequencing after the library was loaded? This question arises because there seems to be no correlation between the initial pore count and the amount of bases obtained.

R2: Available pores refer to flowcell check results, this has been changed in the legend of Figure S1. In fact, there is no correlation between the flowcell check result and the amount of bases sequenced. The flowcell check provides an initial snapshot of the number of active pores, but it doesn't predict the final data yield. This is because pore lifetimes can vary - some pores that are active at the start may degrade quickly. In addition, not all pores generate data at the same rate, DNA quality can limit data production, and sequencing conditions can affect pore activity over time. This has been added in the discussion from line 296 to line 302.

I also have a few structural remarks about the publication:

Line 24: HWM is written instead of HMW in the abbreviation.

Now line 24: corrected

Line 62: There is an unnecessary parenthesis after the word "Us."

Now line 62: corrected

Lines 103/104: The sentence here is duplicated.

Now line 103/104: corrected

Line 499: SQS is written instead of SQK-LSK109.

Now line 512: corrected

Reviewer #2 (Comments for the Author):

The authors present a study aimed at improving the sequencing performance of the OsHV-1 virus, responsible for considerable economic losses in the *Magallana gigas* Pacific oyster industry. It is therefore necessary to produce genomic data enabling the biology of the virus to be studied in detail.

The authors' work is justified by the problems and difficulties of sequencing viruses with large DNA genomes containing repeats and inversions. Shotgun long reads metagenomics is therefore the method of choice for avoiding the need for complex techniques such as TFF, especially when dealing with a non-cultivable virus.

Comparisons of the various kits and bioinformatics algorithms are extremely rigorous, both in methodology and in their description in this article. The evaluation of adaptive sampling is very interesting and promising, which is reassuring because this novelty is not always robust depending on the desired application.

Minor corrections:

Line 24: HMW instead of HWM.

Now line 24: corrected

Lines 104-105: sentence repeated.

Now line 104: corrected

Lines 389-390: please rephrase.

Now line 433-436: corrected

Lines 440-441: Explain why you reduce agitation to 250 rpm and whether you are basing this on experiments.

Now line 486-489: We have reduced the agitation to preserve the HMW DNA as much as possible, this was clarified in the manuscript.

Line 506: Can you describe the homemade scripts in detail?

Response: *We developed all scripts used in this study and these scripts are available here: <https://gitlab.ifremer.fr/lgpmm/ont-bioinformatic-tools-for-oshv-1-genome-characterization> . (also, in line 647).*

Lines 526-527: If you're talking about sequencing depth, it would be preferable to use this term instead of "coverage", using the "X" as the unit.

Now lines 573-576: corrected.

Given that the authors have shotgun metagenomic data in their possession, it would be interesting to have results exploiting this data by providing the DNA virome of the samples.

Response: *We did a viral sequence identification on both the Illumina and Nanopore data with VirSorter2 and CheckV as defined in this protocol [dx.doi.org/10.17504/protocols.io.bwm5pc86](https://doi.org/10.17504/protocols.io.bwm5pc86) . We did find few sequences of dsDNA phages, ssDNA, Iridoviridae and NCDLV viruses. This has been added in the manuscript lines 263-274 414-418 and 578-592.*

Finally, in view of their colleagues' publication establishing a link between telomerase activity and OsHV-1 infection (Dupoué et al. 2024), it would be interesting for the authors to discuss the potential of their developments in this context.

Response: *The aim of our study was to develop an approach that would allow the sequencing and characterisation of the OsHV-1 genome from oyster tissue. Such an approach will undoubtedly allow researchers working on this biological model to study the various aspects of virus-host interactions in greater depth. However, it is beyond the scope of this study to develop the discussion on technological development presented in our manuscript with the work of Dupoué et al.*

Reviewer #3 (Comments for the Author):

Dotto-Maurel and colleagues proposed the paper titled "Evaluation of long-read sequencing for Ostreid herpesvirus type 1 genome characterization from *Magallana gigas* infected tissues". This study is a comparison of different DNA extraction methods and sequencing technologies to reconstruct the OsHV-1 genome. The analysis is interesting and the experimental approach is appropriate, with an adequate number of tested methods. The use of different assemblers to reconstruct the viral genome is also of interest.

The main focus of the study was to develop and test a methodology to effectively enrich for viral DNA over host molecules, in order to propose it as a future diagnostic method for aquaculture. To this end, the use of ONT sequencing over other long-read technologies (e.g. PacBio) is fully justified. However, I guess that briefly mentioning the existence of these other technologies in the introduction might be beneficial, as well as reporting if long-read sequencing has been previously used to investigate OsHV-1 pathogenesis.

Q1: I have a single major concern, whereas several minor points are reported below. The author evaluated the DNA fragmentation running DNA extract on an agarose gel, with a marker with a maximal size of 48.5 kb. Several extraction kits produced HMW DNA located at the very beginning of the run. I guess that using a TapeStation (Agilent) instrument will probably generate more useful results, with the possibility to evaluate also higher sizes (e.g. up to 60 kb), which might differentiate the results.

R1: *While TapeStation analysis would have provided more accurate DNA length results to better compare the kits, we did not have this technology when the study began. However, gel migration still reveals valuable and interesting differences between the seven kits.*

Q2: Moreover, since the averaged length of the ONT reads is shorter than the distribution of DNA extracts (at least as it can be evaluated by the gel picture), the authors should comment this aspect.

R2: *The size discrepancy observed between gel migration and the N50 of the sequenced reads is likely due to two main factors: (i) the library preparation process that could induce DNA fragmentation and (ii) the fact that shorter DNA fragments tend to pass through the sequencing pores with more efficiency than longer DNA fragments inducing a bias toward shorter reads and a decrease of sequenced N50. In fact, for run4, where we did perform a short fragment elimination step, the N50 is larger as small DNA fragments were eliminated. This has been added to the discussion lines 316-320.*

Q3: Can other approaches be used to further enrich for longer DNA fragments before sequencing, thus reducing sequencing efforts and costs?

Q3: *Yes, several methods can enrich for longer DNA fragments prior to sequencing, which can reduce reads of shorter, less informative fragments. In our study, we added a short fragment elimination step in the fourth sequencing run, which enriched for longer fragments (lines 391-402 Figures 2 and 5).*

Alternative methods include gel purification after electrophoresis, although this approach is more labour intensive. Automated size selection systems such as BluePippin can also selectively capture longer fragments. However, these methods add complexity to the workflow and cost around €1,000 for four samples. While these options may increase fragment length, they don't necessarily reduce the overall sequencing effort or cost. SPRIselect Bead for short fragment elimination could be an alternative cost-effective solution, but the loss of total DNA during magnetic bead-based purification implies a large amount of input DNA. Ultimately, the most effective solution is to choose a DNA extraction method that maximises the production of long fragments.

Minor points:

Line 47. The name of the family should be italicized

Now line 47: corrected

Line 77-78. ONT is not the only technology able to produce long reads.

Now line 78-79: corrected

Line 92. Perhaps add "electric"

Now line 92: corrected

Line 96-99. This specification seems trivial and more suitable for a protocol than for an introduction.

Now line 95: corrected

Line 139. What is the definition of "best ratio"?

Now line 140 corrected

Line 165. The origin of the DNA used for Illumina sequencing is, at this point, not defined. Maybe it should be briefly introduced.

Now line 164-166: corrected

Line 275. Correct 13,x into 13.x

Now line 287: corrected

Figure 5. Although this figure contains valuable information, the comparison between conditions is really hard (using a 1-page magnification). Perhaps a different visualization could be considered to allow a more immediate understanding of the differences.

Figure 5 has been modified accordingly.

Methods. Since the flow cells were washed and reused again, it would be interesting to have somewhere reported the number of pores available at the start of each sequencing run and if their decline influence the sequencing readout.

Response: *This information is provided in Figure S1.*

Figure 4. What is the origin of the reads classified as "OsHV-1+oyster"?

Response: *« OsHV-1+M. gigas » reads are reads that align on both OsHV-1 and M. gigas genomes. These reads may be the result of biological chimeras, i.e. virus integrated into the oyster genome, or the result of artefact chimeras due to the ligase used for library preparation. Here we are in the lytic phase and there is no sign of integration associated with a viral pattern, so these reads are most likely artefacts created during library preparation. This has been clarified in the text lines 135 to 138 and 312-314.*

Re: Spectrum02082-24R1 (Evaluation of long-read sequencing for Ostreid herpesvirus type 1 genome characterization from *Magallana gigas* infected tissues)

Dear Dr. Germain Chevignon:

Your manuscript has been accepted, and I am forwarding it to the ASM production staff for publication. Your paper will first be checked to make sure all elements meet the technical requirements. ASM staff will contact you if anything needs to be revised before copyediting and production can begin. Otherwise, you will be notified when your proofs are ready to be viewed.

Sincerely,
Zsolt Toth
Editor
Microbiology Spectrum